# Introduction to the Usage of Open Data from the Large Hadron Collider for Computer Scientists in the Context of Machine Learning

T. Saala[1*] and M. Schott[1]

**1** Institute of Physics, University of Bonn, Germany

⋆ tsaala@uni-bonn.de

## Abstract

Deep learning techniques have evolved rapidly in recent years, significantly impacting various scientific fields, including experimental particle physics. To effectively leverage the latest developments in computer science for particle physics, a strengthened collaboration between computer scientists and physicists is essential. As all machine learning techniques depend on the availability and comprehensibility of extensive data, clear data descriptions and commonly used data formats are prerequisites for successful collaboration. In this study, we converted open data from the Large Hadron Collider, recorded in the ROOT data format commonly used in high-energy physics, to pandas DataFrames, a well-known format in computer science. Additionally, we provide a brief introduction to the data's content and interpretation. This paper aims to serve as a starting point for future interdisciplinary collaborations between computer scientists and physicists, fostering closer ties and facilitating efficient knowledge exchange.

# 1   Introduction

Particle physics aims to uncover the fundamental particles and forces that constitute our universe, addressing deep questions about the nature of matter, energy, space, and time. Modern experiments, like those at the Large Hadron Collider (LHC), generate enormous volumes of data—both real and simulated—that are highly complex, high-dimensional, and feature intricate correlations. These unique characteristics make high-energy physics (HEP) data a compelling and challenging domain for machine learning.

Machine learning has played a crucial role in particle physics for decades, particularly in tasks that involve distinguishing rare signals from overwhelming background noise. Following the breakthrough of deep learning techniques around the 2010s, physicists at the LHC quickly began applying neural networks across many aspects of their experiments. These applications include tasks such as identifying and measuring the properties of particles produced in collisions, classifying entire collision events, and searching for unusual patterns in the data that might hint at previously unknown physical phenomena. In particular, machine learning models are increasingly being used to spot subtle, unexpected features in the data that could reveal new physics beyond the current theoretical framework, the Standard Model - a well-established theory that describes all known fundamental particles and their interactions, except for gravity.

Despite rapid advancements in neural network techniques, it often takes a significant amount of time for the latest developments in the context of computer science to be ported to particle physics applications. One of the main reasons is the complexity of the data structure as well as the availability and access to training data of particle physics experiments for sci-

entists without a background in particle physics. Moreover, for computer scientists, HEP data presents a real-world scenario to test and push the limits of advanced machine learning methods, offering the chance to tackle problems involving massive datasets, noise filtering, and the discovery of rare signals—opportunities that are hard to match in many other domains.

In our lab, we have observed that introducing computer science students to the usage of particle physics data takes a significant amount of time and effort before they can conduct their own research in this - for them unfamiliar - terrain. These notes are intended for computer scientists and researchers interested in developing novel algorithms or testing new approaches using machine learning on Large Hadron Collider (LHC) data. The large general purpose particle detectors at the LHC, ATLAS and CMS, published parts of their data on the CERN Open Data Portal, however, there are two major barriers: first, the LHC data format is based on the ROOT framework, which is largely unknown in the field of computer science. Second, understanding the data structure typically requires a professional background in particle physics.

In this work, we seek to address both barriers. First, we provide a brief explanation for scientists without a dedicated background of particle physics, introducing the observables and the data recorded and analysed at the LHC. Second, we have transformed a substantial portion of the CMS Open Data from the ROOT format to pandas DataFrames, a data format commonly used within the computer science community. The datasets used in this work are based on Run 1 CMS data collected in 2012, which provide a rich and well-documented benchmark for developing and testing machine learning models. These datasets are now also available on bonndata [1] and described in this work.

This paper is structured as follows: the main concepts for the analysis of proton-proton collisions at the LHC are summarized in Section 2. Section 3 explains the concepts of simulation and detector effects, which are essential to understand potential machine learning tasks. A detailed description of the content of the pandas DataFrames is provided in Section 4. The paper concludes with a brief summary, while all technical details regarding the transformation from the ROOT format to pandas DataFrames are included in the appendix.

## 2   A Primer in Experimental Particle Physics

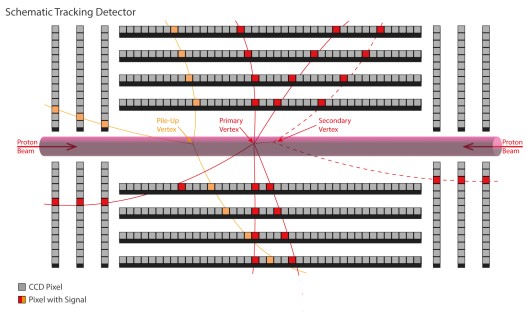

Figure 1: Schematic illustration of a tracking detector including three vertices and several charged particles measured by a pixel detector.

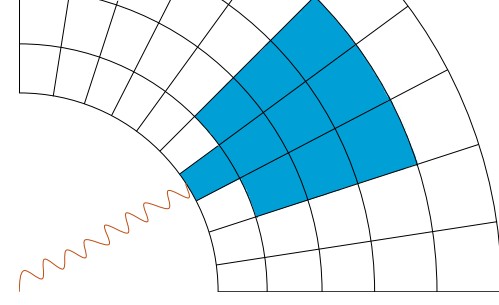

Figure 2: Schematic illustration of an electromagnetic calorimeter and the energy deposits in its cells.

### 2.1   Particles of the Standard Model and Basic Reactions

The Standard Model (SM) is the most successful theory describing the subatomic world, combining the principles of quantum mechanics and special relativity into a quantum field theory. While an in-depth exploration of the Standard Model is beyond the scope of this summary, a

brief overview will provide key insights into the fundamental particles that make up all matter.

Observable matter around us is composed of electrons, protons, and neutrons. Protons and neutrons, in turn, are made of quarks, which exist in two types: up-quarks (u) and down-quarks (d). A proton consists of two up-quarks and one down-quark, resulting in a total electric charge of +1, as the up-quark carries a charge of +2/3 and the down-quark a charge of -1/3. In contrast, neutrons are composed of one up-quark and two down-quarks, giving them a net charge of 0. Thus, the basic building blocks of atomic nuclei are described by up and down quarks.

Quarks are grouped into "doublets" based on their properties, with the up and down quarks forming the first generation. Nature exhibits two additional generations of quarks: the charm (c) and strange (s) quarks form the second generation, and the top (t) and bottom (b) quarks make up the third. Although each generation shares similar characteristics, they differ significantly in mass, with higher-generation quarks being substantially heavier than those of the first.

In addition to quarks, electrons — particles with a charge of -1 — are essential in atomic structure. Electrons are paired in a doublet with the electron-neutrino ($\nu_e$), a neutral particle that is nearly massless. Similarly to quarks, leptons (such as the electron and neutrino) exist in three generations. The second generation consists of the muon ($\mu$) and muon-neutrino ($\nu_\mu$), while the third generation consists of the tau ($\tau$) and tau-neutrino ($\nu_\tau$). These particles also follow a hierarchical mass structure, with higher-generation leptons being significantly heavier than the first-generation electron.

In the Standard Model, neutrinos are assumed to be massless, although we know this is not entirely true. However, this approximation holds for most high-energy calculations in the theory.

Due to the nature of particle interactions, heavier particles are unstable and decay into lighter particles, driven by the principle that systems tend to move towards lower energy states. As a result, the only stable particles in the Standard Model are protons, electrons, and neutrinos (and neutrons when bound within atomic nuclei). These stable particles form the foundation of all matter we observe in the universe.

In addition to all mentioned particles, it turns out that each particle also has its own anti-particle. This is a natural consequence of combining quantum mechanics with special relativity, as described by quantum field theory. Anti-particles have been known for decades and exhibit exactly the same properties as their corresponding particles, i.e. the same mass, but have opposite charges. For example, the anti-particle of the electron, $e^-$ is the positron, $e^+$. The anti-particle of the up-quark is the anti-up quark, which carries an electric charge of $-2/3$ instead of 2/3. The SM predicts that nearly equal amounts of particles and anti-particles exist in the universe and it is one of the main unsolved questions in modern physics, why we are surrounded nearly only by matter particles.

Having discussed the particles that describe the known matter in the universe, we now turn to the forces that govern their interactions. The universe seems to be governed by four fundamental forces: *gravitational, electromagnetic, strong, and weak* forces or interactions. Of these, gravitational force is the weakest, yet it acts over distances on a cosmic scale and governs large-scale structures like planets, stars, and galaxies. However, gravitation is not included in the Standard Model of particle physics, as its effects are negligible at the subatomic level due to its extreme weakness compared to other forces. The Standard Model focuses on the forces that dominate the interactions of elementary particles, where gravity plays no significant role and is therefore not discussed further.

The electromagnetic force governs the interactions between electrically charged particles and is responsible for phenomena such as light, magnetism, and the structure of atoms. It is mediated by photons, the quantum particles of light. The relevant charge for electromagnetic

interactions is the well-known electric charge, with particles carrying either a positive or negative charge. These particles interact by attracting or repelling each other depending on the nature of their charge.

Several important concepts can be illustrated when discussing the electromagnetic force. Consider two electrons in a vacuum separated by a certain distance. According to Coulomb's law, these electrons must repel each other. In a very naive - and absolutely wrong - picture, the repulsion between the two electrons is transmitted by the exchange of photons, akin to two people on skateboards throwing a ball to each other and moving apart as a result. While this analogy might help in explaining repulsion, it fails to account for the attraction expected between oppositely charged particles, such as an electron and a positron. In a more accurate (still wrong and any physicist will rightfully complain) picture, we begin from the field-lines between the two charges. In a quantum field theory - as the name implies - the fields are quantized, and the field-quanta correspond to a (virtual) photon. The effects of repulsion and attraction can then be viewed as the exchange of wave-packages with differently 'signed' amplitudes (this is also not correct, but it is easier to visualize). In this picture, those field-quanta, i.e. photons, can only be exchanged by particles carrying electric charge, since they have an associated electromagnetic field. Only electrically charged particles have field-lines, thus can exchange photons. If a particle does not carry an electric charge, it will be "invisible", consequently not interacting with any electric field-lines at all, thus not interacting with photons.

The strong (nuclear) force binds quarks together to form protons and neutrons and holds atomic nuclei together. This force is mediated by particles called gluons, and the charge associated with the strong force is known as color charge, which comes in three types (red, green, and blue) and their corresponding anti-colors. Quarks carry color charge, and gluons facilitate the force between them. In other words: the quanta of field of the strong interaction are gluons. These gluons can only interact with particles carrying color charge. In fact, this is the main difference between quarks and leptons: quarks carry color charge, hence they can interact with gluons, while all leptons do not have any color charge and hence they do not experience the strong force. An additional complexity arises from the fact that gluons themselves carry color charge. If photons, the mediators of the electromagnetic force, carried electric charge, they would interact with each other. Similarly, the fact that gluons carry color charge means that they can interact with one another. This self-interaction is thought to be the reason why the strong force becomes stronger with distance, a phenomenon known as confinement. Confinement prevents quarks from escaping a bound system, and this is why the strong force has a very short range, playing a significant role only at the nuclear level.

The weak force is responsible for processes such as beta decay in radioactive atoms and the conversion of neutrons into protons. In other words: Without the weak force, the sun would not burn. This force is mediated by massive $W^+$, $W^-$ and Z bosons, where the $W^+$ particle carries a positive electric charge, the $W^-$ particle carries a negative electric charge and the $Z$ boson is electrically neutral. The term "massive" refers to the fact that the W and Z bosons have mass, unlike massless photons and gluons. This might be difficult to comprehend, but imagine mass just as a further property of an object, similar to its electric charge. The weak force has two types of charges, named weak isospin and weak hypercharge. These are less intuitive than electric charge but are essential for distinguishing how particles interact with the W and Z bosons. The weak force affects all fermions (quarks and leptons) and is unique in that it can change one type of particle into another (e.g., turning a down-quark into an up-quark, or changing an electron to a neutrino). The W bosons are responsible for these particle transformations and enable heavier particles to decay into lighter ones. Due to the large mass of the W and Z bosons, the weak force is short-ranged and much weaker than both the electromagnetic and strong forces.

192    Finally, we have to introduce the Higgs Boson. The theory of the SM describes the mass of
193  the W and the Z boson in a naive way, by just writing their mass terms, $m_W$ and $m_Z$ into the
194  formulas which describe the corresponding quantum fields without breaking the predictive
195  power of the theory. A mathematical trick to describe the mass of those two bosons is the
196  introduction of a new field, which we call the Higgs-fields, interacting with the W and Z bosons
197  and 'generating' their mass. This is typically pictured as the W and Z bosons having some form
198  of 'friction' with the Higgs-field and consequently 'slowing down'. While this picture is an
199  oversimplification, it serves as a conceptual aid for understanding the underlying theory. It
200  turns out that this mass generation mechanism would then also give rise to the masses of all
201  quarks and charged leptons. In fact, the theory predicts that the coupling of the Higgs-field
202  is proportional to the mass of a particle. In other words: A particle interacts more with the
203  Higgs-field when its mass is large. Thus, the Higgs boson is simply the first quantized state
204  (or excitation) of the Higgs field. With this, all particles (or more precisely all fields) in the
205  Standard Model have been introduced and are summarized in Table 1.

| Particle | Mass (GeV/$c^2$) | Electric Charge (e) | Lifetime (s) |
|---|---|---|---|
| **Quarks** (Carry Color Charge) | | | |
| Up (u) | 0.0022 | +2/3 | Stable |
| Down (d) | 0.0047 | -1/3 | Stable |
| Charm (c) | 1.27 | +2/3 | $1.1 \times 10^{-12}$ |
| Strange (s) | 0.096 | -1/3 | $1.2 \times 10^{-8}$ |
| Top (t) | 172.76 | +2/3 | $5.4 \times 10^{-25}$ |
| Bottom (b) | 4.18 | -1/3 | $1.5 \times 10^{-12}$ |
| **Leptons** (Carry no Color Charge) | | | |
| Electron (e) | 0.000511 | -1 | Stable |
| Muon ($\mu$) | 0.105 | -1 | $2.2 \times 10^{-6}$ |
| Tau ($\tau$) | 1.776 | -1 | $2.9 \times 10^{-13}$ |
| Electron Neutrino ($\nu_e$) | $< 2.2 \times 10^{-6}$ | 0 | Stable |
| Muon Neutrino ($\nu_\mu$) | $< 0.17$ | 0 | Stable |
| Tau Neutrino ($\nu_\tau$) | $< 18.2$ | 0 | Stable |
| **Gauge Bosons** (Force Carriers) | | | |
| Photon ($\gamma$) | 0 | 0 | Stable |
| W Boson (W$^\pm$) | 80.379 | $\pm 1$ | $3.2 \times 10^{-25}$ |
| Z Boson (Z) | 91.1876 | 0 | $3.0 \times 10^{-25}$ |
| Gluon (g) | 0 | 0 | Stable |
| Higgs (H) | 125.10 | 0 | $1.6 \times 10^{-22}$ |

Table 1: Fundamental Particles of the Standard Model: Masses, Electric charges, and
Lifetimes. The masses of the particles are given in units of GeV/$c^2$, where one GeV
corresponds to $1.783 \cdot 10^{27}$ kg.

206    In everyday life, the electron is the only fundamental matter particle that we can directly
207  observe. Together with protons and neutrons, electrons combine to form atoms, which explain
208  the structure of the periodic table and the foundations of chemistry and biology. As discussed
209  earlier, protons and neutrons are compound objects made up of quarks. However, quarks can
210  also combine to form particles other than protons and neutrons, such as mesons, which consist
211  of one quark and one antiquark. An overview of the most important compound quark systems
212  is given in Table 2. The careful reader will see that the masses of the proton and neutron are
213  roughly equal and about $\approx 1$ GeV/$c^2$ [1]. Since both particles are made up of three quarks, the

---

[1]The unit GeV will be discussed in Section 2.4, but is of secondary importance for the following discussion

214  mass of one quark would therefore be roughly 0.3 GeV. However, a pion has a mass of $\approx 0.14$
215  GeV, but is made up of two quarks. The reason for the difference in masses can be explained
216  by the following: The actual masses of the up and down quarks are very small (Table 1), but
217  when combined, they exchange numerous gluons, forming a bound system with substantial
218  binding energy. From Einstein, we know that energy equals mass. Therefore, the mass of the
219  proton or pion can be attributed almost entirely to binding energy rather than to the mass of
220  the fundamental quarks.

| Particle | Quark Content | Mass (GeV/$c^2$) | Charge (e) | Lifetime (s) |
|---|---|---|---|---|
| Baryons | | | | |
| Proton (p) | $uud$ | 0.938 | +1 | Stable |
| Neutron (n) | $udd$ | 0.939 | 0 | 880 s |
| Pions | | | | |
| Charged Pion ($\pi^{\pm}$) | $\bar{u}d / u\bar{d}$ | 0.13957 | ±1 | $2.6 \times 10^{-8}$ |
| Neutral Pion ($\pi^0$) | $u\bar{u} / d\bar{d}$ | 0.13498 | 0 | $8.4 \times 10^{-17}$ |
| Kaons | | | | |
| Charged Kaon ($K^{\pm}$) | $u\bar{s} / \bar{u}s$ | 0.49367 | ±1 | $1.24 \times 10^{-8}$ |
| Neutral Kaon ($K_L^0$) | $d\bar{s}$ | 0.49761 | 0 | $5.1 \times 10^{-8}$ |
| Neutral Kaon ($K_S^0$) | $d\bar{s}$ | 0.49761 | 0 | $8.9 \times 10^{-11}$ |
| Selected D Mesons | | | | |
| Charged D Meson ($D^{\pm}$) | $c\bar{d} / \bar{c}d$ | 1.869 | ±1 | $1.04 \times 10^{-12}$ |
| Neutral D Meson ($D^0$) | $c\bar{u}$ | 1.865 | 0 | $4.1 \times 10^{-13}$ |
| Selected B Mesons | | | | |
| Charged B Meson ($B^{\pm}$) | $u\bar{b} / \bar{u}b$ | 5.279 | ±1 | $1.64 \times 10^{-12}$ |
| Neutral B Meson ($B^0$) | $d\bar{b}$ | 5.280 | 0 | $1.52 \times 10^{-12}$ |

Table 2: Overview of selected particles that are compound systems of quarks, to-
gether with their quark content and further properties. The masses of the particles
are given in units of GeV/$c^2$, where one GeV corresponds to $1.783 \cdot 10^{27}$ kg.

## 2.2  Feynman Diagrams and Example Processes

222  Proton-proton collisions, such as those observed at high-energy particle accelerators like the
223  Large Hadron Collider (LHC), provide a unique window into the fundamental interactions
224  that govern the subatomic world, i.e. enable testing the Standard Model of Particle Physics.
225  We already discussed that protons are composed of more elementary constituents: quarks
226  and gluons. At high energies, quantum fluctuations in protons become significant, and thus,
227  not only quarks but also antiquarks are found inside them. Thus, at high energies, a proton
228  becomes a very complicated object, as it is composed not only of its three constituent quarks
229  but also of gluon fields and additional quark-antiquark pairs.
230      When two protons collide at high energy, it is therefore not the protons themselves that
231  interact, but rather their constituent quarks and gluons. These interactions can result in a va-
232  riety of complex processes, including the production of new particles such as W and Z bosons,
233  Higgs bosons, and even the top quark. Many of these particles are unstable (Table 1), decay-
234  ing almost immediately into other particles, which can be detected using specialized detectors.
235  The nature of these interactions is described and visualized through Feynman diagrams, which
236  illustrate the initial state (the incoming particles), the interaction process (typically involving
237  intermediate virtual particles), and the final state (the observable products of the interaction).

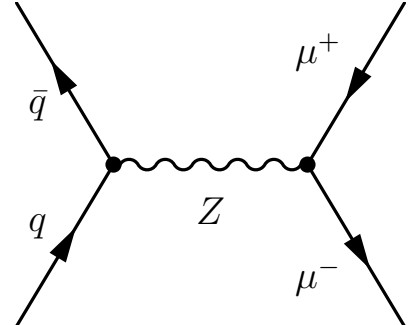

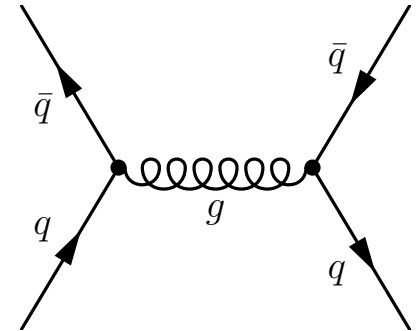

Figure 3: Feynman diagram visualizing the $q\bar{q} \to Z \to \mu^+\mu^-$ process.

Figure 4: Feynman diagram visualizing the $q\bar{q} \to g \to q\bar{q}$ process.

The outcome of a proton-proton collision is inherently statistical, governed by quantum probabilities. The likelihood of specific reactions, and thus the production of particular particles, is determined by quantum field theory. Some processes are significantly more probable than others, resulting in the fact that for each collision, a different set of final particles may emerge, though the probabilities of various outcomes are calculable.

To make this more concrete, we briefly discuss three examples. An important process that can occur in proton-proton collisions is the creation of a Z boson, a neutral carrier of the weak nuclear force. The process can be summarized as follows:

$$q\bar{q} \to Z \to \mu^+\mu^-$$

In this process, illustrated in Figure 3, a quark from one proton and an anti-quark from the colliding proton annihilate to produce a Z boson. The Z boson is highly unstable, with an extremely short lifetime ( $10^{-25}$ seconds). Consequently, it decays almost immediately. In this specific case, the Z boson decays into a pair of oppositely charged muons: a muon and an anti-muon.

The two muons are detectable in the final state, with their tracks and momenta measurable using the detector's tracking systems. The detection of these muons allows experimentalists to infer the presence of the Z boson, even though its short lifetime prevents direct observation.

The most common process in proton-proton collisions involves the production of quark-antiquark pairs through gluon interactions. This process can be expressed as:

$$q\bar{q} \to g \to q\bar{q}$$

In this process, a quark and an anti-quark from the colliding protons annihilate to form a gluon, which then decays into a new quark-antiquark pair (visualized in Figure 4). An important aspect of this process is its high probability relative to other reactions. For example, this type of quark-pair production is about $10^6$ times more likely than the production of a Z boson.

Additionally, the same final state of quarks can result from other processes, such as gluon-gluon fusion:

$$gg \to g \to q\bar{q}$$

In this process, two gluons from the colliding protons interact to form an intermediate gluon, which then decays into a quark-antiquark pair. Since experimental detectors can only observe the final state of particles, distinguishing between these two processes (quark-antiquark annihilation and gluon-gluon fusion) on an event-by-event basis is not possible. Both contribute to the same signature in the detector.

We conclude the discussion with the creation of top quark pairs, as one of the most intriguing processes in high-energy collisions. The process is more complex and can be described as follows:

$$gg \to g \to t\bar{t}$$

In this interaction, two gluons merge to form a highly energetic gluon, which subsequently decays into a top quark and an anti-top quark. The top quark is the heaviest known elementary particle and decays rapidly because of its large mass. The top quark almost exclusively decays via the weak force into a W boson and a bottom quark:

$$t \rightarrow W^+ b$$

The W boson, being unstable itself, decays further into lighter particles. In this example, the W boson decays into an electron and an electron neutrino:

$$W^+ \rightarrow e^+ \nu_e$$

Similarly, the anti-top quark decays into a W boson and an anti-bottom quark:

$$\bar{t} \rightarrow W^- \bar{b}$$

The W boson from this decay can, for instance, decay into a pair of quarks, such as an up quark and a down antiquark:

$$W^- \rightarrow \bar{u} d$$

Thus, the complete process, which is also visualized using a feynmann diagram in Figure 5, can be represented as:

$$g g \rightarrow g \rightarrow t \bar{t} \rightarrow W^+ W^- b \bar{b} \rightarrow e^+ \nu_e b \bar{b} d \bar{u}$$

The final state of this interaction consists of an electron, a neutrino, two b quarks, and two light quarks (up and down). The detection of these final particles allows scientists to reconstruct the original process and analyze the properties of the top quark, W boson, and other particles involved.

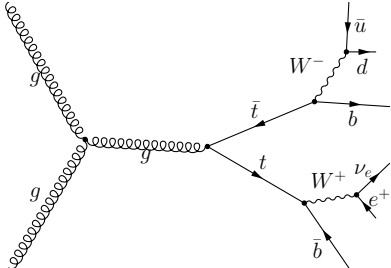

Figure 5: Feynmann diagram visualizing the $g g \rightarrow g \rightarrow t \bar{t} \rightarrow W^+ W^- b \bar{b} \rightarrow e^+ \nu_e b \bar{b} d \bar{u}$ process.

The discussion above simplifies the interaction picture by assuming that only two fundamental particles within the colliding protons interact. However, a single collision between two protons often involves not just one pair of interacting quarks or gluons, but multiple simultaneous interactions among the quarks, antiquarks, and gluons inside the protons. This phenomenon is known as the *underlying event*. In our basic picture of a proton-proton collision, we typically consider the primary interaction between a quark and an antiquark, or between two gluons, which can lead to the production of interesting new particles such as W and Z bosons, Higgs bosons, or top quarks. However, due to the composite nature of protons, other quarks and gluons from the same protons can also engage in separate interactions. These secondary interactions may involve lower-energy exchanges between particles that do not produce exotic or heavy particles but, nonetheless, contribute to the overall energy and particle multiplicity in the final state. Fortunately, new particle production processes often

result in final state particles with significantly higher energies, making them easier to detect and isolate from the softer, low-energy products of secondary interactions.

In addition to the underlying event, high-energy collider experiments must contend with another complication known under the name *pile-up*. Pile-up occurs when many protons collide simultaneously during a single event in the detector. This effect arises because protons in the LHC are not accelerated as single individual particles, but in bunches of protons. Each bunch is approximately 7.5 cm long and contains up to $10^{11}$ protons. Those bunches of protons are then brought to collision. Out of these $10^{11}$ protons, only a fraction actually collide and all other protons just continue their travel in the accelerator. The reason for colliding multiple protons in each bunch is the tiny probability of any individual proton-proton collision to produce an interesting physics process. Therefore, to increase the likelihood of observing rare physics processes, particle accelerators collide multiple protons in each collision event. Hence, particle detectors do not only record one proton-proton collision, but several. For example, during the 2012 data-taking period, between 5 and 40 collisions occurred simultaneously. Those additional proton-proton collisions are called *pile-up*. Since each recorded event contains the superimposed outcomes of several proton-proton collisions, experimentalists must disentangle the products of the interaction of interest from the products of the other simultaneous collisions.

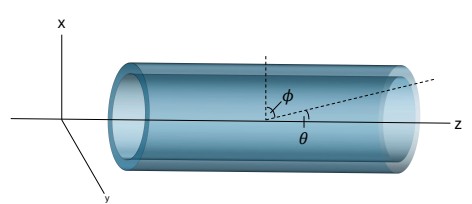

Figure 6: Polar coordinates $\theta$ and $\phi$ represented in the transverse plane of the detector.

Figure 7: Polar coordinate $\phi$ represented in the xy-plane.

## 2.3 Particle Detectors

As discussed previously, the only stable particles at the end of a particle collision are electrons, positrons, neutrinos, protons, neutrons and photons. However, there are several other particles with an average lifetime long enough to allow them to travel several meters or more before they decay. These include many hadrons (Table 2) as well as muons. The primary goal of a typical particle detector, such as those used in high-energy physics experiments at the LHC, is to measure the momentum vector, the energy, as well as the origin of all final-state particles produced around the collision point. These detectors are constructed in a cylindrical shape surrounding the interaction region. The central region is referred to as the barrel detector while the two ends of the cylinder are called the end-caps. The actual particle collisions occur in the center. Figure 10 shows a generic particle detector at the LHC from different perspectives, while Figures 8 and 9 display actual images of the two largest detectors at the LHC: ATLAS [2] and CMS [3]. As the protons are delivered by beam pipes to the collision point, holes are needed on both sides to allow the beam passage. The detector can be schematically divided into several concentric layers, each optimized to detect specific particles and measure their properties. These layers include the inner detector, the electromagnetic calorimeter, the hadronic calorimeter, and the muon chambers.

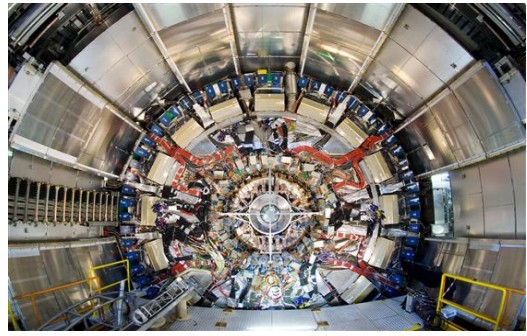

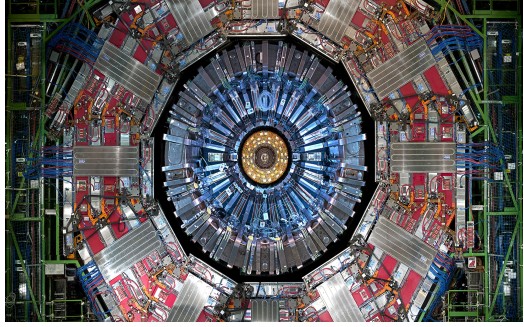

Figure 8: Picture of the ATLAS Experiment at CERN [4].

Figure 9: Picture of the CMS Experiment at CERN [5].

Before discussing these individual detector components, the two main physical principles which allow for an energy and momentum measurement are introduced. The basic idea behind measuring the momentum of a charged particle relies on its motion within a magnetic field: When charged particles transverse a magnetic field, they are bent and the bending radius depends on their momentum. By reconstructing the trajectory of charged particles in a magnetic field, one can therefore determine both their momentum and their electric charge. Hence, each particle detector at the LHC involves a large magnetic field. It is important to note that the actual bending radius of particles that are produced at the LHC is large, given their huge energies, and they can be approximated as straight lines. This also implies that the relative resolution of the measured momenta, i.e. the relative precision of the measurement, worsens with higher momenta, as the trajectories get more and more straight and the difference to a bent line becomes smaller and smaller.

While one can measure the momentum of charged particles, this cannot be done for neutral particles. However, the energy of electrons, photons, and hadrons can be determined through calorimetric measurements. In simple terms, these particles interact with the detector materials and deposit their energy in complex processes, which in turn can be measured. Several things are different compared to the momentum measurement: the relative precision improves with increasing particle energy; the measurement is destructive, meaning the particle is fully absorbed during the process; and to determine the particle's trajectory and origin point, it is typically assumed that the particle originated from a specific reference point in the collision, known as the primary vertex—the location where the initial proton-proton collision occurs.

These features of momentum and energy measurements define the general layout of a particle detector at the LHC: The innermost layer around the interaction point is equipped with a tracking detector in a large magnetic field. This is followed by the so-called electromagnetic and hadronic calorimeters and the muon system as the outermost layer, as illustrated in Figure 10, and detailed in the following:

- **Inner Detector (ID)**: The inner detector typically starts from 5 cm away from the collision point around the beam pipes and extends up to 50 cm. It is designed to measure the transverse momentum $p_T$, i.e. the momentum transverse to the magnetic field which is typically perpendicular to the beam-axis, and charge of all charged particles. The inner detector can be seen as several layers of CCD-cameras, as they are used in smartphones. Each charged particle traversing a pixel of the CCD-camera leaves a signal. When combining those pixel-informations for several layers, the trajectory of the charged particle can be reconstructed (Figure 1). The reconstructed trajectory can then be used to infer the curvature and thus the charge and the momentum. Moreover, the impact parameter as a measure of the origin of the track can be reconstructed.

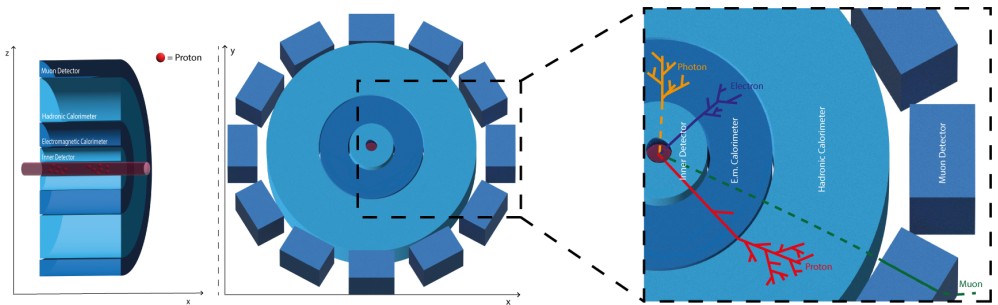

Figure 10: Basic detector layout of an LHC detector with all its sub-detector systems (left) and basic particle identification (right).

- **Electromagnetic Calorimeter (ECAL)**: The electromagnetic calorimeter extends from approximately $r = 1.5$ m to $r = 2.0$ m in typical detectors and is built such that all electrons and photons deposit their full energy there, i.e. they are "absorbed" when transversing the ECAL. The calorimeter is divided into many cells, each measuring the energy that was deposited in it. Once electrons and photons enter the ECAL, they generate an *electromagnetic shower*, which is a process in which electrons and photons produce more and more electrons and photons in an avalanche-like effect. The energy of this shower is measured in the cells. By adding up the energies of all cells that can be "clustered" together (i.e. are nearby), the total energy of the original electrons and photons can be determined. This processes is illustrated in Figure 2. Photons and electrons leave a characteristic shape of energy clusters in the calorimeter, however, one cannot distinguish them by the cluster distribution itself. Those energy cluster distributions are typically described by certain shower-shape variables, which for example reflect the length or the width of the energy distribution in the calorimeter.

- **Hadronic Calorimeter (HCAL)**: Beyond the electromagnetic calorimeter lies the hadronic calorimeter, which typically extends from $r = 2.0$ m to $r = 3.0$ m. The HCAL is designed to measure the energy of hadrons, such as protons, neutrons, and pions, which interact via the strong force. Although hadrons may lose some energy in the electromagnetic calorimeter, they deposit the majority of their energy in the HCAL through hadronic shower processes. While the underlying physics processes are different for electromagnetic showers, the avalanche-like showers and shapes are also present, however, significantly larger. The HCAL is also critical for measuring particle jets, which are groups of hadrons traveling in the same direction. Importantly, it can detect both charged and neutral hadrons, whereas the inner detector can only track charged particles. Hence the reconstruction of particle jet typically relies on information from the ID and all the calorimeter information, i.e. the ECAL and the HCAL.

- **Muon System (MS)**: The only particles that make it beyond the hadronic calorimeter, ignoring neutrinos for now, are muons. They leave a track in the inner detector and lose only a very small amount of energy in the calorimeter system. The muon system is therefore typically also made of tracking detectors, where the bending of muon tracks is measured and an independent determination of their transverse momentum and their corresponding impact parameters can be conducted.

The sub-detector systems are highly specialized and record a vast amount of data per collision event. In fact, the rate of collision events at the LHC is yielding billions of proton-proton

collision per second at each of the LHC detectors. The resulting data rates at the particle detectors would be far too large to be recorded and stored. To solve this problem, each detector has a dedicated *trigger system* that helps select and record interesting or potentially important events from the vast number of proton-proton collisions, by combining very fast detectors and algorithms that assess if an event meets certain criteria. For example, it is always interesting when a highly energetic electron or muon with a transverse momentum above 30 GeV is produced in an event. Additionally, it could be interesting to have two muons in the event, where each of them has a minimal $p_T$ of 10 GeV. It could also be interesting to have an event with two jets with an energy above 50 GeV. By requiring those event characteristics, most of the collision events are not recorded, but only those collision events, where potentially interesting physics processes have occurred. Different selection criteria correspond to different *Trigger Requirements*. Those trigger requirements are chosen, such that the overall recorded event rate is within the technological limits of the data acquisition system. Since each trigger requirement introduces a certain selection bias, the trigger requirements are typically set as minimal as possible. To allow for studies in a completely unbiased way, a certain rate of events are recorded, which do not fulfill a predefined trigger requirement (fire a trigger).

Before discussing what exactly is measured and used for the consequent data analysis, some specific notations need to be introduced.

## 2.4   Notations and Four-Vectors

One of the most famous equations in physics is Einstein's formula for the equivalence of mass and energy:

$$E = mc^2$$

This equation, however, is a simplified form of a more general relationship that accounts for both the energy and momentum of a particle. The full formula is:

$$E^2 = (\vec{p} \cdot c)^2 + (m \cdot c^2)^2$$

In this equation, $E$ represents the total energy of the particle, $\vec{p}$ is the particle's momentum, $m$ is the rest mass of the particle (the mass when it is not moving), and $c$ is the speed of light. When the particle is at rest ($\vec{p} = 0$), the formula reduces to the well-known $E = mc^2$, which describes the energy that is intrinsic to the particle due to its mass.

In high-energy physics, it is common practice to use a system of units called *natural units*, where the speed of light is defined to be 1, i.e. $c = 1$. This simplifies many calculations since the speed of light no longer explicitly appears in the equations. The total energy formula then becomes:

$$E^2 = \vec{p}^2 + m^2$$

This equation relates the energy of a particle to its momentum and mass. For particles moving at high speeds, meaning that their velocity is close to the speed of light, the momentum can become much larger than the rest mass, and their energy is dominated by the momentum term. In these cases, if $E \gg m$, the equation simplifies to:

$$E \approx |\vec{p}|$$

Thus, for high-energy particles, the energy and momentum are approximately equivalent, and the mass can often be neglected when considering the final state particles in high-energy collisions. However, this assumption breaks down for heavy particles, such as the W boson, top quark, or Z boson, where the rest mass is comparable to the total energy.

In everyday life, masses of objects should be measured in kilograms ($kg$), momentum should be measured in $kg \cdot m \cdot s^{-1}$, and energy should be measured in $kg \cdot m^2 \cdot s^{-2}$. These units are impractical for the sub-atomic world, since the mass of a proton is $m_p = 1.6727 \cdot 10^{-27}$ kg. It is much more convenient to use the energy unit of electron volts (eV), where one $eV$ corresponds to the energy that a single electron acquires when flying through a potential difference of 1 Volt. In this unit, the mass can then be expressed as 0.938 GeV/c$^2$, i.e. Giga ($10^9$) electron Volt, divided by $c^2$. Using natural units, the mass of the proton is therefore $m_p$=0.938 GeV$\approx$ 1 GeV.

To better illustrate the relationship between energy, momentum, and mass, we will discuss two examples. The fundamental principle in all physical processes, including collisions of fundamental particles, is the conservation of energy and momentum. As previously discussed, a Z boson, with a mass of approximately 91 GeV/c$^2$ (i.e., in natural units, this is equivalent to 91 GeV), can decay into two muons (a muon and an anti-muon). In this decay, the total energy of the Z boson must be conserved and shared between the two final-state muons. Since the mass of each muon is around 100 MeV (0.1 GeV), this mass is negligible compared to the total energy of the Z boson.

Since the Z boson has 91 GeV of energy and its decay products are two muons, we can assume that each muon receives approximately half of the total energy, i.e., about 45.5 GeV. Since the muon mass is negligible compared to this energy, the momentum of each muon can be approximated as equal to its energy:

$$E_\mu \approx |\vec{p}_\mu| \approx 45.5 \, \text{GeV}$$

Thus, each of the two muons will carry approximately 45.5 GeV of energy and momentum.

As a second example, we consider the decay of a top-quark, with a mass of approximately 172 GeV/$c^2$. When a top quark decays, it typically decays into a W boson and a bottom quark (b-quark). For simplicity, we assume that the energy of the top quark is equally distributed between the W boson and the b-quark, then each would receive approximately 86 GeV of energy.

The mass of the b-quark is relatively small compared to this energy, so its momentum can be approximated as equal to its energy:

$$E_b \approx |\vec{p}_b| \approx 86 \, \text{GeV}$$

However, the W boson has a significant mass of approximately 80 GeV/$c^2$, which is similar to the 86 GeV of energy it receives from the top quark. In this case, the W boson's momentum will be smaller than its energy, because a larger fraction of its energy is tied up in its rest mass. However, the W boson decays further into lighter particles, e.g. one muon and one muon neutrino. These particles will then have the full available energy of 86 GeV, and will ultimately have energies (and momenta) of roughly 40 GeV. These approximations are not completely accurate, as several mass and *phase-space* effects must also be taken into account. However, they provide a first estimate of what can be expected.

Having introduced the relationship between energy and mass, we now turn to discussing what can be measured with a particle detector. First, a coordinate system must be defined. The origin of the coordinate system (0,0,0) is defined at the center of the detector. The z-axis is placed along the beam-line, i.e. along the direction in which the protons enter the detector. The $y-$direction points upwards, while the $x-direction$ is defined to be orthogonal to $y$ and $z$. The $x-$ and $y-$axis define a transverse plane, i.e. transverse to the beam line. The angle in the transverse plane, starting from the $x$-axis is described by $\phi$, the angle between the z-axis and the $radius$ is named $\theta$ (Figure 7). In practice, the angle $\theta$ is expressed with a quantity

named pseudo-rapidity[2] $\eta$, which is defined by

$$\eta = -ln(tan(\theta/2)).$$

Thus, there exists a unique relationship between $\theta$ and $\eta$: $\eta = 0$ corresponds to an angle of 90°, $\eta = 1.0$ corresponds approximately to 45°, and $\eta = 2.5$ corresponds to roughly 10°. A typical LHC particle detector can cover the full region of $\phi = [-\pi, \pi]$ and a region in $\eta$ of $\eta = [-4, 4]$.

With the definition of a coordinate system, we can define the transverse component of the momentum of a particle as

$$\vec{p}_T = (p_x, p_y) = (sin\phi \cdot p_T, cos\phi \cdot p_T)$$

with

$$|p_T| = \sqrt{(p_x^2 + p_y^2)}$$

Since the particle detector measures not only the transverse momentum, but also the trajectory of a particle or the position of the energy cluster in the calorimeter, one gets a direction measurement of the angles $\phi$ and $\theta$, i.e. the pseudo-rapidity. Once those angles and the absolute value of transverse momentum, $p_T$, are known, one can calculate all three components of the momentum vector, via

$$p_x = sin\phi \cdot p_T, \qquad p_y = cos\phi \cdot p_T, \qquad p_z = tan\theta \cdot p_T$$

In case of massless particles, or particles whose rest mass is significantly smaller than their momentum, the momentum is equivalent to the energy and we can define energy values in three dimensions. At this point, this seems absurd since energy is a scalar quantity, however, its usefulness will become apparent in section 2.5.

$$E_x = sin\phi \cdot E_T, \qquad E_y = cos\phi \cdot E_T, \qquad E_z = tan\theta \cdot E_T$$

The information of the energy of a particle, $E$, and its momentum vector $\vec{p}$ is combined in special relativity in a four-vector object, defined as

$$p = (E, p_x, p_y, p_z).$$

The scalar product of one four-vector $p^1$ and a second four-vector $p^2$ is defined as

$$p^1 . p^2 = E^1 \cdot E^2 - p_x^1 \cdot p_x^2 - p_y^1 \cdot p_y^2 - p_z^1 \cdot p_z^2 = (m^{12})^2$$

and results in the so-called invariant mass $m^{12}$ of the two four-vectors. The minus signs after the energy are motivated by special relativity, which is beyond the scope of this introduction. To interpret this expression, it is illustrative to calculate the scalar product of a four-vector with itself, yielding

$$p . p = E \cdot E - p_x \cdot p_x - p_y \cdot p_y - p_z \cdot p_z = E^2 - |p|^2 = m^2,$$

i.e. the rest mass of the particle. Now consider the decay of a Z boson into two muons and assume the momentum vector of the two muons has been measured, i.e. we know $p_x$, $p_y$ and $p_z$ for the positively and negatively charged muon. Since the muon is nearly massless,

---

[2]The reason for this choice lies in the special relativity and the easier interpretation or particle properties for physicists; most light particles at the LHC are produced in forward direction and the number of particles per unit $\eta$ stays roughly constant.

515 we know that the energy of one muon must be $E = \sqrt{p_x^2 + p_y^2 + p_z^2}$, hence we can define the
516 four-vector for each muon separately. By adding those two four-vectors, we get the kinematics
517 of the mother particle, i.e. the four-vector of the Z boson. By taking the scalar product of this
518 four-vector, we obtain the rest mass (invariant mass) of the $Z$ boson.
519     In summary, the relevant quantities are four-vectors of final state particles, since they allow
520 for a reconstruction of the properties of the intermediate states of a particle collision.

## 2.5   Primary Objects, Particle Identification and Derived Observables

522 The raw data from energy clusters in the electromagnetic and hadronic calorimeters, along
523 with track information from the inner detector and muon system, can be combined to extract
524 additional information about the particles created in the collision. The basic idea is illus-
525 trated in Figure 10: electrons are expected to leave a track in the inner detector, which can
526 be matched to an energy deposit in the electromagnetic calorimeter system, while the photon
527 just leaves an energy cluster in the electromagnetic calorimeter but no track can be associated.
528 Similarly, protons would leave all their energy in a cluster in the hadronic calorimeter, where
529 a reconstructed track from the inner detector should point to. In contrast, a neutron is not
530 electrically charged, thus leaving no track in the inner detector instead just an energy deposit
531 in the hadronic calorimeter. Muons could be identified by tracks in the inner detector that can
532 be matched to tracks in the muon system. Clearly, the underlying concepts are more complex
533 and some of those aspects are discussed in more detail in the following.

534 **Particle Flow Objects:**   In a first step, one tries to associate all tracks in the inner detector
535 and all clusters in the calorimeter systems. The matching is typically done using the $\eta$ and $\phi$
536 variables as well as the reconstructed momenta and energies. The result is a so-called parti-
537 cle flow object, containing not only basic variables $p_T$, $\eta$, $\phi$ and charge, but also information
538 about the mass of the particle, the associated reconstructed energies of the clusters in the
539 electromagnetic and hadronic calorimeters, as well as the likelihood of the matching. Further-
540 more, its origin, i.e. the vertex, is saved. Clearly, some information might not be available,
541 e.g. when an association between a cluster and a track is not feasible, since - in this case -
542 either the energy information or the vertex are unavailable. Particle Flow Objects are the basis
543 for all subsequent objects, as they represent all primary objects that have been measured by
544 particle detector systems.

545 **Electrons:**   Naively, each track that can be associated with an ECAL cluster, while containing
546 no corresponding energy deposit in the HCAL could be identified as an electron. Unfortu-
547 nately, some hadrons can also mimic a similar signature. In order to lower the probability that
548 a hadron is falsely identified as an electron, the very specific shower shapes of electrons in the
549 calorimeter are used to further separate electron from hadron processes. Sadly, this classifica-
550 tion is still not ideal. Not all real electrons are reconstructed and identified as electrons and
551 some reconstructed electron candidates are still caused by hadrons. The more stringent the
552 selection criteria on the shower-shape variables, the smaller the fake-rate, although conversely
553 resulting in a reduced identification efficiency for electrons and photons.
554     Since the energy measurement of a calorimeter improves with higher energies, one typi-
555 cally takes the measurement of the energy from the calorimeter, while the measurements of $\eta$
556 and $\phi$ are taken from the inner detector. Another reason for this is that electrons can radiate
557 photons when transversing the material of the inner detector via the Bremsstrahlung process.
558 By emitting Bremsstrahlung photons, the charged particles lose energy, and the curvature of
559 the track increases, consequently making it difficult to correctly estimate their initial momen-
560 tum. However, the energy of these Bremsstrahlung photons is measured by the calorimeter,

561 and is therefore included in the energy measurement.

562 **Photons:**  The signatures of photons are very similar to those of electrons, except that no
563 track in the inner detector can be associated with the reconstructed electromagnetic cluster.
564 However, some photons interact with the material of the inner detector system and split up
565 to an electron positron pair, which then leaves tracks in the inner detector system. Hence
566 a reconstructed photon also contains information on whether such a process has happened
567 previously. Similarly to electrons, hadrons can also fake photon signatures. Therefore, the
568 shape information of the deposited energies in the electromagnetic calorimeter is used for
569 improving the identification, i.e. lowering the probability that a hadron is incorrectly identified
570 as a photon, but still keeping the efficiency of identifying a photon relatively high.

571 **Muons:**  Muons are all particle flow objects containing a corresponding track in the muon
572 system. The likelihood that other particles mimic a muon signature is extremely small, im-
573 plying that all reconstructed muon candidates are actually caused by muons. Although the
574 fake-rate is small, the track association, as well as the track reconstruction, is not perfect,
575 implying certain inefficiencies.

576 **Vertices:**  In addition to the energy and momentum of particles, one can also determine the
577 positions where particles are produced or have been colliding. These positions are called
578 vertex and are described by three coordinates $(v_x, v_y, v_z)$. The vertex is reconstructed using
579 the reconstructed trajectory of a single charged particle by the inner detector. The minimal
580 distance of a trajectory to the z-axis is described by the so-called impact parameters, $d_0$ and $z_0$
581 for the transverse plane and the $xy-$plane respectively, as illustrated in Figures 6 and 7. By
582 combining the information of several trajectories, a common origin can be determined, which
583 in turn is defined as a vertex.
584      Typically, several vertices are reconstructed in each proton-proton collision. The so-called
585 primary vertex corresponds to the position of the initial proton-proton collision, where the
586 most interesting physics reaction took place, e.g. the creation or decay of a heavy particle,
587 such as the Higgs boson or a top-quark. The primary vertex is typically defined as the vertex
588 having the largest number of - or the highest energetic - tracks associated to it. Secondary
589 vertices are usually found close to the primary vertex and typically have two or three tracks
590 originating from them. They stem from decays of particles that were produced in the primary
591 vertex but are long-lived enough that they can travel a few mm before decaying further. Typical
592 examples are tau-leptons or B-mesons, i.e. bound systems which contain one b-quark.
593      In addition to the primary vertex and associated secondary vertices, there are also vertices
594 originating from the other proton-proton collisions that are recorded in the same event. Those
595 vertices are labeled as *pile-up* vertices. The reconstruction pile-up vertices can be used to mit-
596 igate the effect of pile-up events on the reconstructed quantities. For example, reconstructed
597 energy clusters in the calorimeter, which have an associated track that stems from a pile-up
598 vertex, can be ignored and excluded from further analysis. Nevertheless, pile-up will always
599 reduce the detector performance, since neutral particles stemming from pile-up collisions will
600 lead to a signature in the calorimeter systems, but cannot be directly identified due to not
601 leaving a track in the ID.

602 **Jets:**  Quarks and gluons in the final state hadronize, yielding a number of charged and neu-
603 tral hadrons, flying and spreading out as a group of particles along the direction of the original
604 quark or gluon. These hadrons deposit their energies in the form of energy-clusters in the ECAL
605 and the HCAL systems, as well as tracks of the charged hadrons in the inner detector. This
606 concept is illustrated in Figure 11.

In order to identify which energy deposits correspond to the signature of the original quark or gluon, special algorithms have been developed. The most prominent one is the *Anti-$K_T$* algorithm, which can be applied to all particle flow objects in the $\eta, \phi$-plane. Particle flow objects that are grouped together following certain rules, are called a particle jet. A particle jet can therefore have energy, momentum, and a direction described by $\eta$ and $\phi$. Moreover, we can add up relativistic momenta of all particle flow objects, and subsequently define the invariant mass for the particle-jet.

The number and properties of particle jets that have been found in a collision event depend on a parameter $R$, which needs to be defined at the beginning of the Anti-$K_T$ algorithm. $R$ is defined between two particles as

$$R = \sqrt{\Delta\eta^2 + \Delta\phi^2},$$

i.e. it can be interpreted as a distance in the $\eta, \phi$-plane. Its maximal value is typically set to 0.4 or 0.6 in the Anti-$K_T$, which governs the size of the particle jet and defines at which distance other particle flow objects are no longer considered.

Jets originating from quarks and gluons yield very similar signatures in the calorimeters, however, machine learning techniques can be used in order to find subtle differences, allowing for a certain degree of separation. The separation between typical jets and those originating from b-quarks (and to some extent from *c*-quarks) is significantly simpler, as those jets typically have their origin with a certain displacement from the primary vertex, i.e., have reconstructed secondary vertices.

**Tau Leptons:** Similarly to b- and c-quarks, $\tau$ leptons also have a very typical decay signature, which can be reconstructed directly using tracks from the inner detector system. Since more advanced algorithms are required, we will not discuss this in detail. As a result of those algorithms, we can reconstruct the kinematics of the tau-leptons, i.e. $p_T, \eta, \phi$ and its charge, albeit, with a rather small efficiency and a significant fake-rate.

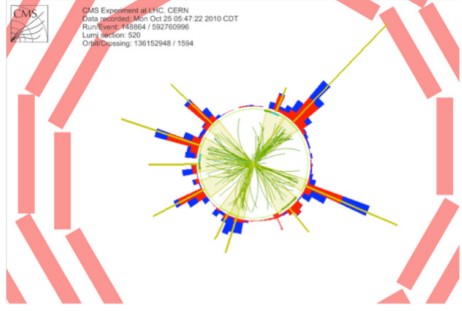

Figure 11: Exemplary depiction of a jet reconstruction. This image was taken from a news article: [6], by the CMS Collaboration.

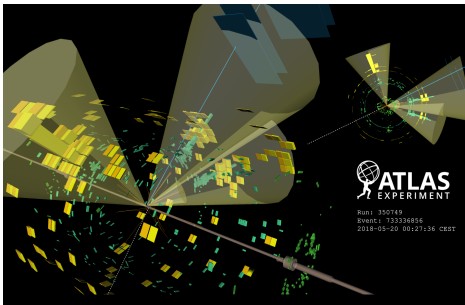

Figure 12: Event Display of a recorded proton-proton collision by ATLAS, showing a candidate of a produced top-quark pair that decays into one muon and four particle jets, while containing missing energy. This picture was taken from [7].

**Neutrinos and Missing Transverse Energy:** The interaction probability of neutrinos with normal matter is vanishingly small, making it nearly impossible to detect them directly, as they pass through the detector without leaving a measurable signal. However, since neutrinos carry energy, their presence can be inferred indirectly through the principle of energy and momentum conservation. When summing up the energies of all visible particles in a collision

event, any imbalance in energy can indicate the presence of neutrinos or other undetected particles.

From the LHC settings, we know that each proton has an energy of 7000 GeV in the z-direction and nearly zero transverse momentum in the x- and y-directions. However, the exact fraction of the momentum carried by quarks and gluons (partons) inside each proton is unknown. For instance, one quark might carry 30% of the proton's momentum, while another carries only 10%. Consequently, we do not have precise knowledge of the partonic momenta in the z-direction before the collision. However, we know that the total momentum of the protons in the transverse plane (x- and y-directions) is zero before the collision. Consequently, through momentum conservation, the sum of the transverse momenta of all final-state particles must also be zero. Mathematically, this condition can be expressed as:

$$|\sum_i \vec{E}_T^i| = |\sum_i (E_x^i, E_y^i)| = 0$$

Any deviation from this expected zero sum in the transverse plane is referred to as *missing transverse energy* (typically denoted as $\vec{E}_T^{\text{miss}}$ or $MET$). This quantity serves as a proxy for the transverse energy of neutrinos or other undetected particles that escaped the detector system without depositing energy. $MET$ is a key observable in events involving weakly interacting particles, such as neutrinos or hypothetical dark matter candidates.

**Isolation:** Electrons and muons originating directly from the decay of heavy particles, e.g. the $Z$ boson, appear isolated in the inner detector. Isolation means that no (or only a few other) tracks or particle flow objects are in close proximity. This is very different for electrons and muons produced during the creation of particle jets which have significant activity around them. To quantify this, several isolation variables can be defined, e.g. as the sum of all transverse momenta around the muon or electron within a $\Delta R < 0.2$ cone radius. This can then be used to place a requirement on the isolation of a reconstructed electron or muon, $iso_{pt} = \sum_i^{alltracks} p_T^i < \text{maximal value } GeV$. Sometimes, not a definite maximal value, but a relative definition, i.e. $iso_{pt}/p_T^{lepton}$ is used for the isolation definition. The reason for this is simple: electrons and muon signatures in the detector stemming from particle jets are much more likely than those that come from resonance decays. Hence, one typically requires *isolated* leptons when studying processes that involve direct decays into leptons.

## 3 Simulation, Detector Response and Machine Learning Tasks

### 3.1 Simulation of Proton-Proton Collisions

Our current understanding of nature is based on a Quantum Field Theory named Standard Model of Particle Physics, which was shortly introduced in section 2. The theory allows for impressive predictions on a sub-atomic level, in particular, it can be used to predict what happens when one collides two protons like at the LHC. Since the Standard Model is based on quantum physics, only probabilities for certain reactions can be given. For example, it is very likely that two particle jets will emerge in a proton-proton collision, but it takes more than $10^9$ collisions to produce a single Z boson. Consequently, a vast amount of collisions need to be analyzed to discover and study the few events where something intriguing occurs.

The Standard Model is formulated elegantly on a mathematical level, but in order to allow for predictions, several approximations and rather complex computational approaches are necessary. Predictions on proton-proton collisions are carried out by programs called *Event*

Generators. Since Monte Carlo methods are used within those programs, the underlying simulations are also referred to as *MC Simulations*. Several different programs are currently used, each differing in the approximations made during their calculations. They all share the ability to predict the outcome of a given number of proton-proton collisions, i.e., they internally simulate probabilistic outcomes to determine what could happen in a collision and repeat this process for subsequent collisions. Given that the probability for something interesting happening is so small, one typically defines what kind of collision events should be produced, e.g. one can define that one Z boson should be produced every time. The outcome of the simulation of one proton-proton collision via an event generator is then the kinematics of all stable particles after the collision. Since each collision is a random process, the amount, as well as the kinematics of all stable particles will be different for the next proton-proton collision. The information of the outcome of the simulation is called generator level, or MC truth level.

## 3.2   Simulation of Particle Detectors and Reconstruction of Objects

If we were a god-like figure, then we would see nature at this MC truth level, i.e. we would know exactly what happened in each collision event: which quarks and gluons have been interacting, what their energy was, what they created, and which particles have been produced in their decay. However, we are just physicists and hence we can only observe what can be measured by our detector, which also needs to be simulated.

The starting point are simulated stable particles for a given event after the proton-proton collision. Their path through the actual particle detector as well as the induced electronic signals have to be simulated. At this point, a simulated event is treated exactly like an event of a real collision and all reconstruction algorithms are applied. The electronic signals in the inner detector are used to reconstruct trajectories of charged particles, the deposited energy signals in the calorimeter are used to reconstruct particle jets and so forth. The reconstructed objects at this stage are called *detector level* or *reconstruction level* objects.

The difference between *generator level* and *detector level* is an important point, which we want to illustrate with a simple example. Assume a proton-proton collision where one electron is expected to be produced. We know the kinematics of this electron from the event generator program, i.e. we know its kinematics on *generator level*. The path of the electron is then simulated in a second step through the detector, where its interaction with the inner detector and the ECAL is estimated. Based on this information, we then try to reconstruct the kinematics of the electron. Clearly, our measurements of the track momentum and track direction or the energy in the calorimeter are not perfect. Hence, the reconstructed electron is close but not identical to the original. Sometimes it might occur that we do not reconstruct an electron at all, as it might fail certain identification criteria or just flies through an un-instrumented part of the detector.

Therefore, there is a significant difference between the particles at *generator level* and *detector or reconstruction level*, i.e. after the effect of the detector on the measurement. This difference does not apply only to electrons, but to all measured quantities. In particular, the difference between the two 'layers' grows larger, when the detector resolution gets worse.

In real collisions, we can only see quantities on *detector or reconstruction level* but never at generator level. Hence, it is crucial to have simulated samples of various processes in proton-proton collisions, as they are needed to interpret what actually can be measured. In fact, simulated MC samples of processes are used in three different ways: First, they are used such that experimental physicists know what to expect when looking for a certain process. For example, let us assume we want to select collision events where a *W* boson has been created and decayed further to one electron and one neutrino. On detector level, we know from simulations what transverse momentum distributions the electron might have, what its isolation properties are, and how the reconstructed missing transverse energy distribution $\vec{E}_T^{\mathrm{miss}}$ looks

like. This allows defining selection criteria on data, e.g. by requiring a minimal transverse momentum of the electron to be $p_T >$ 25 GeV, an isolation $iso_{pt}/p_T^{lepton} < 0.1$ and a minimal missing transverse energy of $\vec{E}_T^{\text{miss}} > 30$ GeV. When having such a selection, one will realize that other physics processes also pass this selection, which are not from the $pp \rightarrow W \rightarrow e\nu_e$ process. An example would be the process $pp \rightarrow Z \rightarrow e^+e^-$, i.e. the creation of a Z boson and its decay into one electron and one positron. When one electron leaves the detector undetected, for example through the beam-pipe, this electron would be interpreted on reconstruction level as missing transverse energy, hence also passing the above described signal selection.

The second purpose of MC samples is, therefore, to estimate how often such background processes pass the signal selection and also to optimize the signal selection criteria, such that the ratio of signal over background events is improved. It is important to note that one can never definitely conclude of any observed event signature about the underlying process. Instead, one calculates probabilities for various processes that yield a certain observed events characteristics. This is the reason why typically large statistics in data are necessary to draw statistically significant conclusions.

Thirdly, MC simulated samples can be used to extract physics parameters. To illustrate this, we discuss a possible determination of the mass of the Z boson: We can simulate the expected reconstructed invariant mass distribution of Z boson events, which decay into two electrons, and vary the assumed mass of the Z boson in the simulation. In a second step, the reconstructed invariant mass distribution of Z boson candidate events in data is compared to the different MC predicted distributions for various Z boson mass approximations. The best-fitting approximation can then be used to determine the fundamental mass parameter in the theory.

The simulation of proton-proton collisions does not fully resemble reality, since the modeling of the proton-proton collisions suffers from theory uncertainties but also because the detector response, i.e. the simulated signals within the detector are not described correctly. The first kind of mismodeling can be tested by varying the theoretical assumptions made during the event generation, or by simply using two different event generator programs. The second kind of mismodeling, i.e. those on the detector level, are corrected by additional smearing corrections or reweighting of events. Assume, for example, that the simulation predicts that the measured muon momentum is always larger by 5% compared to reality, then one merely *rescales* each reconstructed muon momentum in simulated samples by 1/1.05. If the reconstruction efficiency of one muon is in simulation 95% but in reality only 92%, then the events are weighted by a factor 0.92/0.95. Those corrections can be derived by studying processes that are well known, and hence one knows exactly what should be measured in principle. In fact, an enormous effort is put in by the LHC collaborations to derive such *data/MC* correction factors. Without their detailed understanding, no serious data-analysis can be performed in high energy physics.

# 4 Event Records

The most important concepts of LHC collision data have been introduced, hence the actual data structure can be discussed. In a first step, we present a simple example of a $TopAnti - Top$ pair produced in proton-proton collisions, which immediately decay semi-leptonically and discuss how such events are stored in principle. A semi-leptonic decay typically results in both a hadron, and a lepton in the final state. We then delve into the full detailed event record information and summarize all available samples of the CMS Open Data in the pandas DataFrame format.

### 4.1 Simple Example

We store extensive information about each collision in the form of variables, totaling 121 in quantity. In this context, a single collision can also be called an event. These saved variables either retain information about the underlying event itself, or about the physical objects encountered within the event. In the following example, we will focus mostly on variables containing information about the objects encountered in an event. This category can be further divided into the individual objects, namely Particle Flow (PF) objects, vertices, Monte Carlo Truth (MCTruth) objects, electrons, muons, taus, photons, as well as jets. Here, we will focus on muons and PF objects.

Now, depending on the collision-type, as well as the event itself, the amount of these objects measured by the detector can vary significantly. In the DataFrames this is represented by the variables prefixed with "n", such as nPF, which for each event contain the amount of PF objects observed. If we consider three randomly selected events of the collisions described previously, one could see results as depicted in Table 3.

| Amount of certain objects observed. | | | | | | | | |
|---|---|---|---|---|---|---|---|---|
| Event | nEle | nMuon | nTau | nPhoton | nPF | nVertex | nMcTruth | nJets |
| 1 | 0 | 3 | 64 | 0 | 1046 | 11 | 774 | 64 |
| 2 | 1 | 4 | 98 | 1 | 1820 | 26 | 591 | 98 |
| 3 | 0 | 7 | 96 | 0 | 1524 | 18 | 885 | 96 |

Table 3: Amount of certain objects observed in three distinct events of proton-proton collisions.

On a larger scale, using 10000 events instead of just three, the distributions of how many events contain certain amounts of objects can be visualized through histograms. As can be inferred by looking at Table 3, there tend to be considerably more PF objects compared to other objects such as Muons or Electrons. This is due to the fact that Particle Flow objects act as a kind of super-object, which contains all other sub-objects. Another factor that can influence the amount of certain objects encountered in a dataset is the underlying particle collision or decay. For example, when considering a Z-Boson decaying to two electrons, one would expect most events in this sample to have exactly 2 electrons.

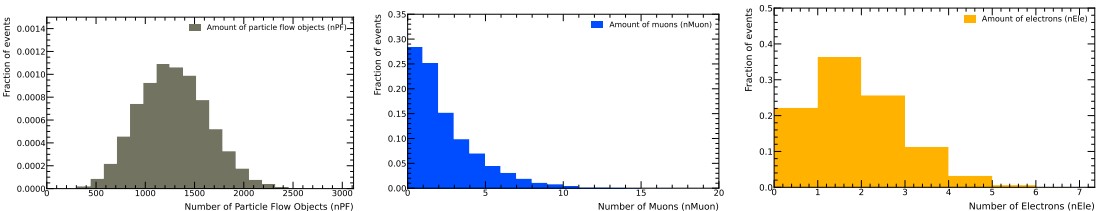

Figure 13: Histograms visualizing the distributions of how many events contain which amount of PF objects (left), muons (middle), and electrons (right) over 10000 events.

This idea is visualized - for PF Objects, Muons, and Electrons - in the histograms found in Figure 13 , where the middle histogram shows the amount of muons encountered in a Top Anti-Top pair decaying semi-leptonically, and the right one shows the amount of electrons encountered in the same decay. When analyzing these distributions, one can observe that the amounts of muons and electrons encountered in the events do not exactly match the expected results. For the considered decay, one would expect to detect exactly one lepton per event, as both muons and electrons are leptons, one would therefore expect to only encounter either 0 or exactly 1 muon / electron per event. These errors stem mostly from low energy objects that get falsely detected. When looking at the histograms in Figure 14, one can see that there are indeed many low energy muons and electrons for both datasets.

This error can be offset by applying cutoffs on the measured transverse momentum of the respective objects, for example, by applying a cutoff of considering only muons and electron which have a momentum of above 5 GeV, the distributions better represent the expected results. These histograms can be seen in Figure 15.

As mentioned previously, the variables stored in the DataFrames are categorized into event-information and object-information. To distinctly associate object-information variables with specific objects, a naming convention has been applied. Here, the prefix is the general datatype the variable is represented as within an event: $vec$ for arrays, $f$ for floats, and $n$ for the integers describing how many instances of an object are observed within the event. The infix then names the precise object, for example, $Muon$, $PF$, or $Jet$. Then, separated by an underscore, the suffix represents which variables information is saved, for example $Eta$ for $\eta$ or $Phi$ for $\phi$. Putting all of this together, the information about the transverse momenta of the encountered

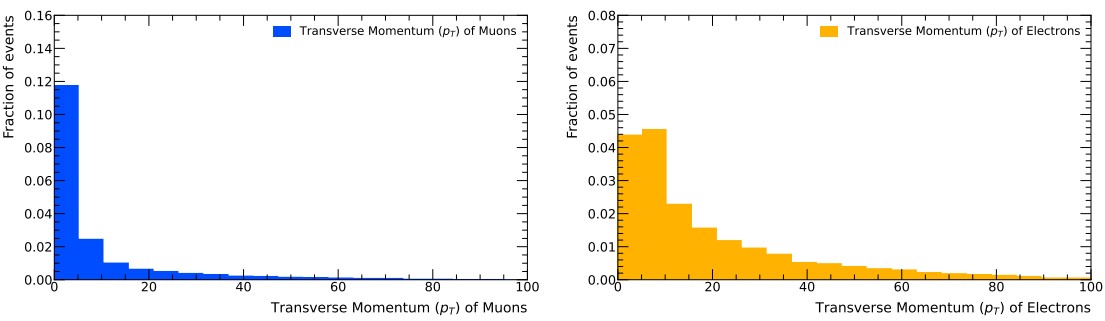

Figure 14: Histograms visualizing the distributions of how many muons (left) / electrons (right) have specific Transverse Momentum over 10000 events.

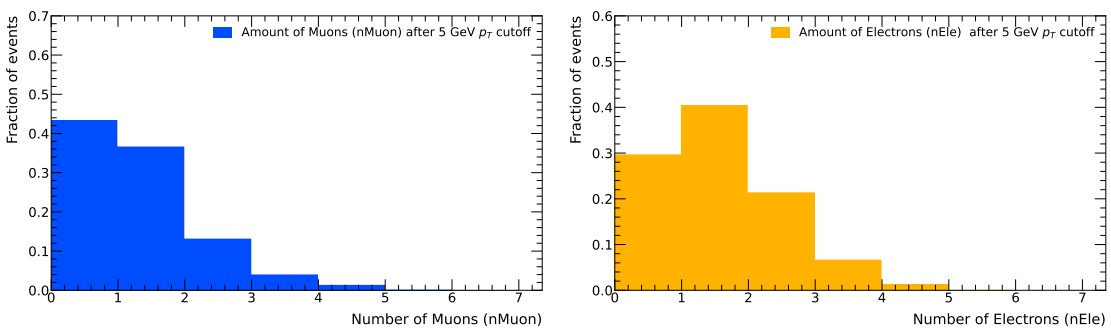

Figure 15: Histograms visualizing the distributions of how many events contain which amount of muons (left) / electrons (right) over 10000 events, after applying a 5 GeV cutoff on the Transverse Momenta of the encountered objects.

muons would be stored in the array *vecMuon_PT*. A full list of the variables using this naming convention and their descriptions can be seen in Table 7. For the event-information, no real naming convention is applied, however, the relevant variables and their descriptions can be found in Table 13.

The length of the variables represented as arrays (prefixed with *vec*) depends on how many of the related object were encountered in the given event. In Table 3 for example, the first event contains three muons, hence all array variables for the muon are of length three in this event. Here, a value $v$ at index $i$ in the respective array represents the value for the variable of the $i$-th object in the event. Some variables are saved for multiple objects, such as their transverse momentum, $\eta$, and $\phi$. Their values, however, vary depending not only on the event they belong to, but also which object they are related to. Consider for now the variables $p_T$, $\eta$, and $\phi$. For the same three events listed in Table 3, their values for muons, electrons, and PF objects are shown in Table 4. These three variables are in fact stored for each physical object, as well as the MET, as they uniquely describe the direction, as well as the momentum the respective object was either measured, or in the case of Monte Carlo Truth objects, how it truly would be if the detector were perfect.

Increasing the sample size to 10000 events instead of three, the distribution of variables for certain objects can once again be visualized using histograms. Histograms representing said distributions for PF objects on their transverse momentum and their polar-coordinates $\eta$ and $\phi$ can be seen in Figure 16.

A popular way of leveraging high energy physics data in deep learning approaches is encoding them as images. These images can then be used in powerful Convolutional Neural Networks. To this end, a straightforward encoding of the available data in the DataFrames is

**Transverse momentum, $\eta$, and $\phi$ for multiple objects.**

**PF Objects**

| Event | nPF | $pT_0$ | $\eta_0$ | $\phi_0$ | ... | $pT_{1046}$ | $\eta_{1046}$ | $\phi_{1046}$ | ... | $pT_{1819}$ | $\eta_{1819}$ | $\phi_{1819}$ |
|---|---|---|---|---|---|---|---|---|---|---|---|---|
| 1 | 1046 | 0.4 | -1.3 | 1.2 | ... | - | - | - | ... | - | - | - |
| 2 | 1820 | 2.8 | -1.2 | -1.3 | ... | 0.2 | -3.6 | -0.4 | ... | 0.2 | -0.9 | -1.4 |
| 3 | 1524 | 0.9 | -1.9 | -0.9 | ... | 0.2 | 3.8 | 0.9 | ... | - | - | - |

**Muons**

| Event | nMuon | $pT_0$ | $\eta_0$ | $\phi_0$ | ... | $pT_2$ | $\eta_2$ | $\phi_2$ | ... | $pT_6$ | $\eta_6$ | $\phi_6$ |
|---|---|---|---|---|---|---|---|---|---|---|---|---|
| 1 | 3 | 70.2 | -2.4 | -2.8 | ... | 1.2 | -1.9 | -2.0 | ... | - | - | - |
| 2 | 4 | 8.3 | 1.53 | -0.0 | ... | 19.7 | 0.9 | 1.0 | ... | - | - | - |
| 3 | 7 | 11.1 | -0.3 | 2.8 | ... | 2.1 | -1.9 | 0.3 | ... | 0.9 | -1.9 | 2.8 |

**Electrons**

| Event | nEle | $pT_0$ | $\eta_0$ | $\phi_0$ | $pT_1$ | $\eta_1$ | $\phi_1$ | ... | $pT_6$ | $\eta_6$ | $\phi_6$ | ... |
|---|---|---|---|---|---|---|---|---|---|---|---|---|
| 1 | 0 | - | - | - | - | - | - | ... | - | - | - | ... |
| 2 | 1 | - | - | - | 2.8 | -1.3 | -1.2 | ... | - | - | - | ... |
| 3 | 0 | - | - | - | - | - | - | ... | - | - | - | ... |

Table 4: Schematic representation of how variables are saved depending on the number of objects observed in an event, based on PF objects, Muons, and Electrons in a $ZZ \to 4\mu$ decay.

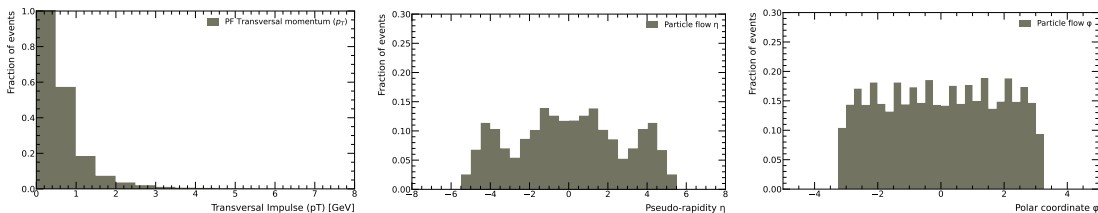

Figure 16: Histogram visualizing the distributions of $p_T$ (left) $\eta$ (middle) and $\phi$ (right) over 10000 events for PF objects.

to use the three variables discussed previously: transverse momentum, $\eta$, and $\phi$. The images can then be created for each event by discretizing the $\eta$ and $\phi$ ranges as bins and then summing up transverse momenta at the corresponding bin overlap. A resulting greyscale image for the PF objects of a single event can be seen in Figure 17.

## 4.2 Detailed Information

In this section, a complete list of variables within the pandas DataFrames will be provided in tabular format, as well as a brief introduction into the objects. Multiple Tables are presented, categorized by the type of variable they concern. Specifically, the variables are here categorized into general variables (Table 13), muon variables (Table 7), vertex object variables (Table 9), electron variables (Table 6), tau variables (Table 8), photon variables (Table 10), Monte Carlo truth variables (Table 11), jet variables (Table 12), as well as PF object variables (Table 5). Each table is constructed as follows: within the first column the name of the variable is stated. In the second column, its datatype is given, i.e Integer, Float, or any numpy ndarray. Lastly,

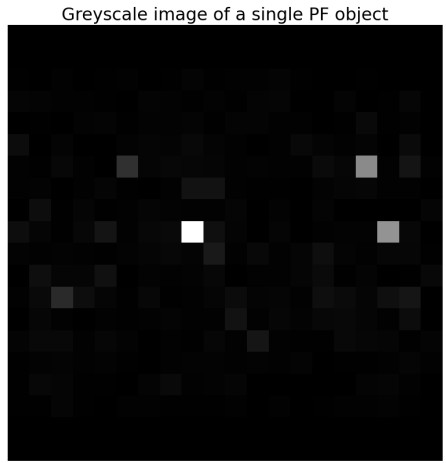

Figure 17: Greyscale image of transverse momentum of a single events PF objects, ranging over $\eta$ and $\phi$.

the third column provides a brief description of what the variable entails physically.

### 4.2.1 Particle Flow Objects

Particle Flow objects are special since they contain all of the other physical objects, such as muons, electrons, photons, and tauons. Additionally, these PF objects are used to construct high-level objects, such as the PF jets and the missing transverse momentum. For the PF objects we store the respective $p_T$, $\eta$, and $\phi$ values, as is done for every physical object in this context. This is done since these three quantities can be used to uniquely define the point in space the respective object was measured within the detector, as well as which momentum the object had in the transverse plane during its measurement. A variable unique to PF objects is PfType, which stores information about the particle IDs (pgdId) of the objects. This ID defines which specific object the PF object is, e.g. an object having a pgdId of 11 is an electron, while a pgdId of -11 would be linked to a positron. A complete list of the variables saved for the PF objects can be found in Table 5.

| Particle Flow (PF) Objects | | |
|---|---|---|
| **Name** | **Type** | **Description** |
| nPF | Integer | Number of PF objects in the respective event. |
| vecPF_PT | Float-ndarray | Array of the transverse momenta ($p_T$) of the PF objects in the respective event. |
| vecPF_Eta | Float-ndarray | Array of the pseudorapidities ($\eta$) of the PF objects in the respective event. |
| vecPF_Phi | Float-ndarray | Array of the azimuthal angles ($\phi$) of the PF objects in the respective event. |
| vecPF_E | Float-ndarray | Array of the energies of the PF objects in the respective event. |
| vecPF_Q | Float-ndarray | Array of the electric charges of the PF objects in the respective event. |
| vecPF_Mass | Float-ndarray | Array of the invariant masses of the PF objects in the respective event. |
| vecPF_PfType | Integer-ndarray | Array containing the particle types (PDG IDs) of the PF objects in the respective event. |
| vecPF_EcalE | Float-ndarray | Array of the energies measured in the Electromagnetic Calorimeter (ECAL) for the PF objects in the respective event. |
| vecPF_HcalE | Float-ndarray | Array of the energies measured in the Hadronic Calorimeter (HCAL) for the PF objects in the respective event. |
| vecPF_ndof | Float-ndarray | Array of the number of degrees of freedom (ndof) of the track fit associated with the PF objects (where applicable) in the respective event. |
| vecPF_Chi2 | Float-ndarray | Array of the chi-squared ($\chi^2$) values from the track fit associated with the PF objects (where applicable) in the respective event. |
| vecPF_pvId | Integer-ndarray | Array of the primary vertex IDs associated with the PF objects in the respective event. |
| vecPF_X | Float-ndarray | Array of the $x$ coordinates of the closest vertex associated with each PF object in the respective event. |
| vecPF_Y | Float-ndarray | Array of the $y$ coordinates of the closest vertex associated with each PF object in the respective event. |
| vecPF_Z | Float-ndarray | Array of the $z$ coordinates of the closest vertex associated with each PF object in the respective event. |
| vecPF_JetNum | Integer-ndarray | Array indicating the jet index to which each PF object belongs (–1 if the PF object does not belong to any jet). This is used to link PF jets to their constituent particles. |

Table 5: Description of the variables associated with Particle Flow objects contained in the pandas DataFrames.

### 4.2.2 Electrons

For electrons, in addition to saving their $p_T$, $\eta$, and $\phi$ we also save their charge, denoted by Q. For electrons, this can be either -1 for normal electrons, or +1 for anti-electrons, also called positrons. A full list of the variables concerning electrons can be found in Table 6.

| Electrons | | |
|---|---|---|
| **Name** | **Type** | **Description** |
| nEle | Integer | Amount of electrons measured in the respective Event. |
| vecEle_PT | Float-ndarray | Array of the Transverse Momenta ($p_T$) of the measured electrons in the respective event. |
| vecEle_Eta | Float-ndarray | Array of the pseudo-rapidities $\eta$ of the measured electrons in the respective event. |
| vecEle_Phi | Float-ndarray | Array of the polar coordinates $\phi$ of the measured electrons in the respective event. |
| vecEle_Q | Float-ndarray | Array of the charges of the measured electrons in the respective event. -1 for electrons and +1 for positron |
| vecEle_TrkIso03 | Float-ndarray | Array of the summed up Transverse Momenta ($p_T$) of the isolation of tracks within a radius of 0.3 for the measured electrons. |
| vecEle_EcalIso03 | Float-ndarray | Array of the summed up Electromagnetic Calorimeter Energies (EcalE) of the isolation of tracks within a radius of 0.3 for the measured electrons. |
| vecEle_HcalIso03 | Float-ndarray | Array of the summed up Hadronic Calorimeter Energies (HcalE) of the isolation of tracks within a radius of 0.3 for the measured electrons. |
| vecEle_D0 | Float-ndarray | Array of the impact parameters (d) in xy direction in the respective event. |
| vecEle_Dz | Float-ndarray | Array of the impact parameters (d) in z direction in the respective event. |

Table 6: Description of the variables associated to electrons contained in the pandas DataFrames.

### 4.2.3 Muons

For muons, we also save their $p_T$, $\eta$ and $\phi$, and their charge. Additionally, an array of the errors associated to the Transverse Momenta is saved under the variable PTErr, as well as saving information about measured standalone muons in the respective events, specifically about the $p_T$, $\eta$ and $\phi$, called StaEta, StaPhi, and StaPt respectively. Again, a full list of variables pertaining to muons can be seen in Table 7.

| Muons | | |
|---|---|---|
| **Name** | **Type** | **Description** |
| nMuon | Integer | Amount of muons measured in the respective Event. |
| vecMuon_PT | Float-ndarray | Array of the Transverse Momenta ($p_T$) of the measured muons in the respective event. |
| vecMuon_Eta | Float-ndarray | Array of the pseudo-rapidities $\eta$ of the measured muons in the respective event. |
| vecMuon_Phi | Float-ndarray | Array of the polar coordinate $\phi$ of the measured muons in the respective event. |
| vecMuon_PTErr | Float-ndarray | Array of the errors associated to the Transverse Momenta ($p_T$) of the measured muons in the respective event. |
| vecMuon_Q | Float-ndarray | Array of the charges of the measured muons in the respective event. -1 for muon and +1 for anti-muon |
| vecMuon_StaPt | Float-ndarray | Array of the Transverse Momenta ($p_T$) of the measured standalone muons in the respective event. |
| vecMuon_StaEta | Float-ndarray | Array of the pseudo-radidity $\eta$ of the measured standalone muons in the respective event. |
| vecMuon_StaPhi | Float-ndarray | Array of the polar coordinates $\phi$ of the measured standalone muons in the respective event. |
| vecMuon_TrkIso03 | Float-ndarray | Array of the summed up Transverse Momenta ($p_T$) of the isolation of tracks within a radius of 0.3 for the measured muons. |
| vecMuon_EcalIso03 | Float-ndarray | Array of the summed up Electromagnetic Calorimeter Energies (EcalE) of the isolation of tracks within a radius of 0.3 for the measured muons. |
| vecMuon_HcalIso03 | Float-ndarray | Array of the summed up Hadronic Calorimeter Energies (HcalE) of the isolation of tracks within a radius of 0.3 for the measured muons. |

Table 7: Description of the variables associated to muons contained in the pandas DataFrames.

### 4.2.4 Tauons

For tauons, in addition to saving the usual variables as for the previously discussed objects, we also save information about the raw isolation of the objects in several variables, namely RawIso3Hits, RawIsoMVA3oldDMwoLT, RawIsoMVA3newDMwoLT, and RawIsoMVA3newDMwLT. The full description of tauons variables can be seen in Table 8.

| Tauons | | |
|--------|------|-------------|
| **Name** | **Type** | **Description** |
| nTau | Integer | Amount of tauons measured in the respective Event. |
| vecTau_PT | Float-ndarray | Array of the Transverse Momenta ($p_T$) of the measured tauons in the respective event. |
| vecTau_Eta | Float-ndarray | Array of the pseudo-rapidities $\eta$ of the measured tauons in the respective event. |
| vecTau_Phi | Float-ndarray | Array of the polar coordinates $\phi$ of the measured tauons in the respective event. |
| vecTau_Q | Float-ndarray | Array of the charges of the measured tauons in the respective event. -1 for tauons and +1 for anti-tauons |

Table 8: Description of the variables associated to electrons contained in the pandas DataFrames.

### 4.2.5   Vertex Objects

The position of a proton-proton collision in the detector is defined by three coordinates, called the vertex. The vertex of a single proton-proton collision is estimated by extrapolating the reconstructed particle tracks to a common origin, a process known as vertex reconstruction. At the LHC, where many collisions occur in a single bunch crossing, multiple such vertices can be reconstructed in one event. The primary vertex refers to the collision point of main interest—typically the one with the highest track activity and most likely associated with the hard scatter process being studied. In contrast, pile-up vertices originate from additional, usually lower-energy proton-proton collisions occurring in the same bunch crossing and are reconstructed separately to isolate the primary physics signal. Meanwhile, secondary vertices are displaced from the primary vertex and result from the decay of long-lived particles, such as B hadrons. Identifying and distinguishing between these different types of vertices is essential for accurately reconstructing the event topology and suppressing background contributions. A full description of vertex object variables can be found in Table 9.

### 4.2.6   Photons

Photons introduce several new variables, such as Hovere (read as H over E). This variable stores information about the energy deposited in the Electromagnetic (E) and Hadronic Calorimeters (H) respectively. The full descriptions of the photon variables can be seen in Table 10.

### 4.2.7   Monte Carlo Truth Objects

Monte Carlo Truth objects are another special type of objects. They do not necessarily represent specific physical objects such as leptons or photons, instead they instead represent a collection of multiple such objects. Specifically, Monte Carlo Truth objects represent the respective objects as they would be observed in an ideal detector, or in this case they represent the objects before the detector step (either with an actual or a simulated detector), therefore immediately after the generation of the particles. Again, these Monte Carlo Truth objects store the common variables such as $p_T$, $\eta$, and $\phi$. As was the case in PF objects, MC Truth objects also contain a reference to the particles pgdId. Additionally, we save information about the flavour codes of the primary and secondary mother vertices of the respective particle in the variables Id_1 and Id_2. A full description of all MC Truth object variables can be found in Table 11.

| Vertex Objects | | |
|---|---|---|
| **Name** | **Type** | **Description** |
| nVertex | Integer | Number of reconstructed vertices in the respective event. |
| vecVertex_nTracksfit | Integer-ndarray | Array of the number of tracks used in the fit of each vertex in the respective event. |
| vecVertex_ndof | Float-ndarray | Array of the number of degrees of freedom (ndof) from the vertex fit for each vertex in the respective event. |
| vecVertex_Chi2 | Float-ndarray | Array of the chi-squared ($\chi^2$) values from the vertex fit for each vertex in the respective event. |
| vecVertex_X | Float-ndarray | Array of the $x$ coordinates of the fitted position of each vertex in the respective event. |
| vecVertex_Y | Float-ndarray | Array of the $y$ coordinates of the fitted position of each vertex in the respective event. |
| vecVertex_Z | Float-ndarray | Array of the $z$ coordinates of the fitted position of each vertex in the respective event. |

Table 9: Description of the variables associated with vertex objects contained in the pandas DataFrames.

### 4.2.8 Jets

A jet is a clustering of particles that, by some approximations, go in roughly the same direction within the detector. In this context, the jets here are calculated using the PF objects, hence they are Particle Flow Jets, however, the jet variables containing the prefix "Gen" hold information about so-called generator-level jets (gen jets). These jets are similar to the MC Truth objects in that they pertain to objects that are regarded before the detector was either simulated or really used. As a jet is a clustering of particles, some variables of jets are used to store information about how many different types of particles are in a jet. For example, the variable nParticles stores how many particles are in a jet in total, while the variable nNeutrals stores how many particles in a jet have neutral charge (0). A full description of the variables stored for jet objects can be found in Table 12.

| Jets | | |
|---|---|---|
| **Name** | **Type** | **Description** |
| nJets | Integer | Amount of particle jets in the respective Event. |
| vecJet_PT | Float-ndarray | Array of the Transverse Momenta ($p_T$) of the particle jets in the respective event. |
| vecJet_Eta | Float-ndarray | Array of the pseudo-rapidities $\eta$ of the particle jets in the respective event. |
| vecJet_Phi | Float-ndarray | Array of the polar coordinates $\phi$ of the particle jets in the respective event. |
| vecJet_Q | Float-ndarray | Array of the charges of the particle jets in the respective event. |
| vecJet_Mass | Float-ndarray | Array of the masses of the particle jets in the respective event. |
| vecJet_D0 | Float-ndarray | Array of the impact parameters (d) in xy direction in the respective event. |

| vecJet_Dz | Float-ndarray | Array of the impact parameters (d) in z direction in the respective event. |
|---|---|---|
| vecJet_nCharged | Integer-ndarray | Array of the amount of charged particles for the given particle jet in the respective event. |
| vecJet_nNeutrals | Integer-ndarray | Array of the amount of neutrally charged particles for the given particle jet in the respective event. |
| vecJet_nParticles | Integer-ndarray | Array of the amount of particles for the given particle jet in the respective event. |
| vecJet_Beta | Float-ndarray | Array containing the $\beta$-values of the jet in the respective event (measure of how well the tracks associated with a jet point back to the primary vertex). |
| vecJet_BetaStar | Float-ndarray | Array containing the $\beta^*$-values of the jet in the respective event (measure the contribution of tracks not from the primary vertex). |
| vecJet_dR2Mean | Float-ndarray | Array containing the mean values of the $\Delta R$ distances between jet constituents in the respective event. |
| vecJet_Area | Float-ndarray | Array containing the values for the area of the jet in the R-plane in the respective event. |
| vecJet_Energy | Float-ndarray | Array of the energy of the particle jets in the respective event. |
| vecJet_chEmEnergy | Float-ndarray | Array of the fraction of energy of the charged particle jets being deposited into the electromagnetic calorimeter. |
| vecJet_neuEmEnergy | Float-ndarray | Array of the fraction of energy of the neutral particle jets being deposited into the electromagnetic calorimeter. |
| vecJet_chHadEnergy | Float-ndarray | Array of the fraction of energy of the charged particle jets being deposited into the hadronic calorimeter. |
| vecJet_neuHadEnergy | Float-ndarray | Array of the fraction of energy of the neutral particle jets being deposited into the hadronic calorimeter. |
| vecJet_mcFlavor | Integer-ndarray | Array of the McTruth flavor associated to the jets in the given event. |
| vecJet_GenPT | Float-ndarray | Array of the Transverse Momenta of generator level jets that are matched to the PF jets of a given event. |
| vecJet_GenEta | Float-ndarray | Array of the pseudo-rapidities $\eta$ of generator level jets that are matched to the PF jets of a given event. |
| vecJet_GenPhi | Float-ndarray | Array of the polar coordinates $\phi$ of generator level jets that are matched to the PF jets of a given event. |
| vecJet_GenMass | Float-ndarray | Array of the masses of generator level jets that are matched to the PF jets of a given event. |
| vecJet_flavorMatchPT | Float-ndarray | Array of the Transverse Momenta of jets that are matched by their flavor. |
| vecJet_ID | Integer-ndarray | Array of quality measurements of the jets for a given event. 0 means no quality, 1 loose quality, and 2 tight quality. |
| vecJet_Num | Integer-ndarray | Array of index of jets, in order of decreasing Transverse Momentum of the PF jets in an event. |
| vecJet_MatchIdx | Integer-ndarray | Array referencing which generator level jets belong to which PF jet. Referenced by the jets index, in order of decreasing Transverse Momentum of the generator jets. |
| vecJet_JEC | Float-ndarray | Array of the jet energy correction factors of the PF jets in a given event. |

Table 12: Description of the variables associated to Monte Carlo Truth contained in the pandas DataFrames.

| Photons | | |
|---|---|---|
| **Name** | **Type** | **Description** |
| nPhoton | Integer | Amount of photons measured in the respective Event. |
| vecPhoton_PT | Float-ndarray | Array of the Transverse Momenta ($p_T$) of the measured photons in the respective event. |
| vecPhoton_Eta | Float-ndarray | Array of the pseudo-radidities $\eta$ of the measured photons in the respective event. |
| vecPhoton_Phi | Float-ndarray | Array of the polar coordinates $\phi$ of the measured photons in the respective event. |
| vecPhoton_Hovere | Float-ndarray | Array of the fraction of energy being deposited in the hadronic (h) and electromagnetic (e) calorimeter respectively, per event. |
| vecPhoton_Sthovere | Float-ndarray | Array of the fraction of energy being deposited in the hadronic (h) and electromagnetic (e) calorimeter respectively, per event. |
| vecPhoton_Has-PixelSeed | Boolean-ndarray | Array containing flags if the photon has left a signature in the inner detector in the respective event. |
| vecPhoton_IsConv | Boolean-ndarray | Array containing flags if the photon is converted into one electron and one positron in the respective event. |
| vecPhoton_Pass-ElectronVeto | Boolean-ndarray | Array containing flags if the photon passed the veto, that it is not identified as an electron in the respective event. |

Table 10: Description of the variables associated to photons contained in the pandas DataFrames.

### 4.2.9 General Variables

Lastly, we also store some "general" variables, which contain meta-information about the underlying collision, process, and dataset. Within this category, we store information regarding the missing transverse energy, specifically its transverse momentum $p_T$, as well as $\eta$ and $\phi$. The missing transverse energy stored here is reconstructed using particle flow combines information from all subdetectors to identify and reconstruct individual particles in an event. By classifying particles as charged or neutral and associating tracks with calorimeter deposits, the particle flow algorithm provides a more precise measurement of the momentum imbalance in the transverse plane. This approach improves the resolution and pile-up robustness of the missing transvers energy, especially in events with complex final states.

Additionally, we save trigger information, which are boolean variables stating whether the respective event passed the selection trigger. A full list of general variables stored in the pandas DataFrames can be found in Table 13.

### 4.3 Data and Simulation Samples

A large amount of datasets are available on the CERN Open Data platform. Broadly, one can divide the entirety of the datasets into two categories, datasets coming from real particle collisions measured at the LHC, which we will call data samples, as well as datasets containing simulated samples, called simulation samples.

| Monte Carlo (MC) Truth | | |
|---|---|---|
| **Name** | **Type** | **Description** |
| nMctruth | Integer | Number of MC truth particles (from the generator) in the respective event. |
| vecMctruth_PT | Float-ndarray | Array of the transverse momenta ($p_T$) of the MC truth particles in the respective event. |
| vecMctruth_Eta | Float-ndarray | Array of the pseudo-rapidities $\eta$ of the MC truth particles in the respective event. |
| vecMctruth_Phi | Float-ndarray | Array of the azimuthal angles $\phi$ of the MC truth particles in the respective event. |
| vecMctruth_Mass | Float-ndarray | Array of the masses of the MC truth particles in the respective event. |
| vecMctruth_Mothers.first | Integer-ndarray | Array of indices identifying the first mother particle of each MC truth particle in the respective event. |
| vecMctruth_Mothers.second | Integer-ndarray | Array of indices identifying the second mother particle of each MC truth particle in the respective event. |
| vecMctruth_Id_1 | Integer-ndarray | Array containing the PDG ID of the incident parton from the first proton beam participating in the hard scattering (event-level quantity). |
| vecMctruth_Id_2 | Integer-ndarray | Array containing the PDG ID of the incident parton from the second proton beam participating in the hard scattering (event-level quantity). |
| vecMctruth_X_1 | Float-ndarray | Array containing the fractions of the beams momentum carried by the incident parton from the first proton beam in the hard scattering. |
| vecMctruth_X_2 | Float-ndarray | Array containing the fractions of the beams momentum carried by incident parton from the second proton beam in the hard scattering. |
| vecMctruth_PdgId | Integer-ndarray | Array containing the PDG IDs of the MC truth particles in the respective event. |
| vecMctruth_Status | Integer-ndarray | Array of status codes specifying the generator-level status of the MC truth particles in the respective event. |
| vecMctruth_Y | Float-ndarray | Array of the rapidities of the MC truth particles in the respective event. |

Table 11: Description of the variables associated to MC truth contained in the pandas DataFrames.

## 5 A Simple Data Analysis

In order to give a more concrete feeling of a typical LHC data analysis, we will discuss the cross-section measurements of W boson events in proton-proton collisions at an energy of 8 TeV. This cross-section can be thought of as a measure of the probability that this process happens in a proton-proton collision and can be calculated within the Standard Model. A full cross-section measurement is quite complicated, so we will focus here on one aspect, namely the estimation of the number of signal events in a given data-set. For this, we study the data-sets, summarized

| General Variables | | |
| --- | --- | --- |
| **Name** | **Type** | **Description** |
| nEvent | Integer | Number of the event within a given DataFrame. Between 1 and 10000. |
| runNum | Integer | Run number of the underlying dataset. |
| evtNum | Integer | Number of the event within the underlying CMS dataset. |
| lumisection | Float | Lumisection of the events collision. |
| fMET_PT | Float | Transverse Momentum ($p_T$) of the events Missing Transverse Energy (MET). |
| fMET_Eta | Float | Pseudo-rapidity $\eta$ of the events Missing Transverse Energy (MET). |
| fMET_Phi | Float | Polar coordinate $\phi$ of the events Missing Transverse Energy (MET). |
| HLT_Mu17_Mu8 | Boolean | Flag representing whether the HLT_Mu17_Mu8 trigger has been met (True) or not (False). |
| HLT_Mu24 | Boolean | Flag representing whether the HLT_Mu24 trigger has been met (True) or not (False). |
| HLT_MET120_v | Boolean | Flag representing whether the HLT_MET120_v trigger has been met (True) or not (False). |
| HLT_Ele27 | Boolean | Flag representing whether the HLT_Ele27 trigger has been met (True) or not (False). |
| HLT_HT350 | Boolean | Flag representing whether the HLT_HT350 trigger has been met (True) or not (False). |

Table 13: Description of the general variables contained in the pandas DataFrames.

| Name | DOI | Sim. | Cross Section | Number of Events |
| --- | --- | --- | --- | --- |
| DYToMuMu_M-20_CT10_TuneZ2star_v2_8TeV [8] | 10.7483 /OPENDATA.CMS.QGC3.PTZ9 | Sim. | 1870.4 pb | ~600.000 |
| QCD_Pt-40_doubleEMEnriched_TuneZ2star_8TeV [9] | 10.7483 /OPENDATA.CMS.L4NC.EV0K | Sim. | 1 pb | ~7.300.000 |
| WplusToMuNu_CT10_8TeV [10] | 10.7483 /OPENDATA.CMS.I3N4.AVW3 | Sim. | 6706.74 pb | ~300.000 |
| **Name** | **DOI** | **Data** | **Int. Luminosity** | **Number of Events** |
| SingleMu [11] | 10.7483 /OPENDATA.CMS.IYVQ.1J0W | Data | 26278.9 pb$^{-1}$ | ~8.700.000 |
| DoubleMu [12] | 10.7483 /OPENDATA.CMS.RZ34.QR6N | Data | 8385.73 pb$^{-1}$ | ~4.700.000 |

Table 14: Summary of CMS Open Data simulated and real datasets transformed to pandas DataFrames that were used in the examplary data analysis.

in Table 14 and the reader is encouraged to implement the following discussion in Python. A GitHub Repository to follow the code and reproduce results can be found here [13] and the required datasets can be downloaded on Hugging Face [14].

   We start by looking at one decay channel of the *W* boson into one muon and one muon-

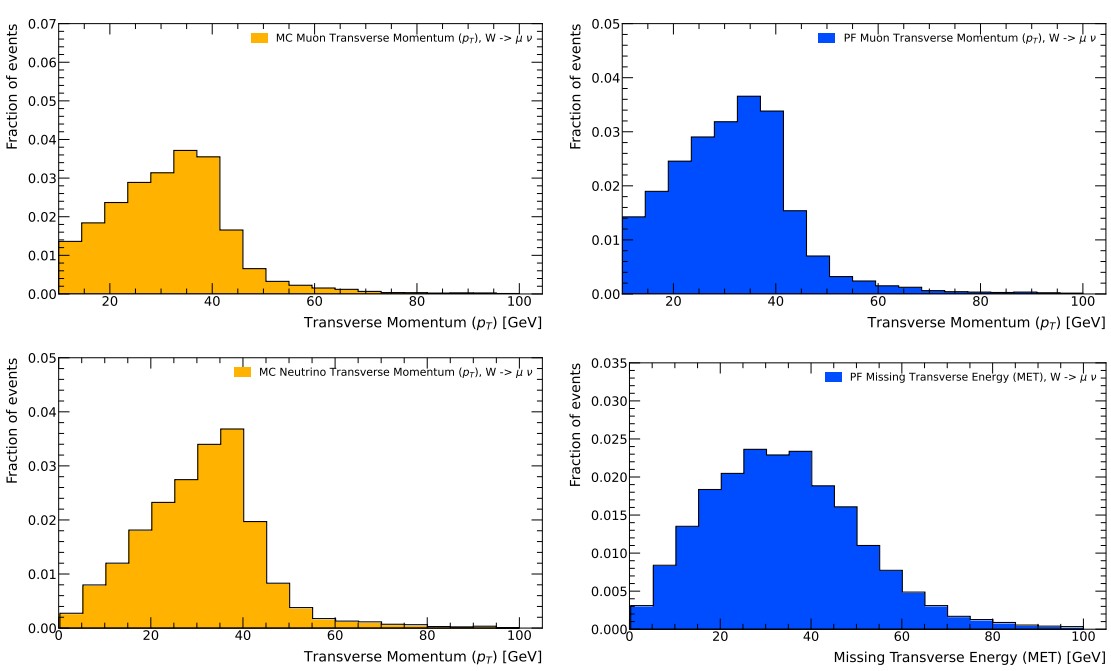

Figure 18: Upper row: MC Truth $p_T$ Muon left, MC Reco $p_T$ muon Right. Lower row: MC Truth $p_T$ Neutrino left, ETMiss Reco Right.

neutrino, e.g. the process $pp \rightarrow W^{\pm} \rightarrow \mu^{\pm} \nu$. In a first step, one needs to define certain signal selection criteria. The final state already implies that we expect one muon in the detector, however, we do not yet know which transverse momentum distribution we should expect. Figure 18 shows the $p_T$ distribution of all MC Truth muons that stem from the decay of a $W$ boson on the top-left side, while the bottom-left side shows the reconstructed $p_T$ distribution of all reconstructed muons for all events. The differences in these two distributions are mainly due to the limited detector resolution, i.e. the fact that the measured $p_T$ is always a bit different than the true $p_T$. Similarly, we can compare the $p_T$ distribution of the sum of the neutrinos at MC truth level as well as the reconstructed missing transverse energy, shown in the same figure on the right side. The differences are larger here, as the detector resolution for this observable is significantly poorer. It can already be concluded that the signal must include exactly one reconstructed muon and a certain minimal value of $E_T^{Miss}$

However, the distributions of the signal sample alone will not allow drawing any conclusions on possible selection criteria on the kinematics of the decay muon and the decay neutrino. For this, possible background processes have also to be studied. In this example, the production of particle jets, for example in the reaction $pp \rightarrow Z \rightarrow q\bar{q}$, as well as the decay of a Z boson in two muons in the process $pp \rightarrow Z \rightarrow \mu^+\mu^-$ need to be considered. The first process is typically called multi-jet production. As discussed in Section 2, muons can also be produced during the hadronization process. Hence, one muon might simply be produced within one particle jet. Given the very limited detector resolution on $E_T^{Miss}$, some events which do not even have a neutrino in the final state might still be reconstructed with a significant $E_T^{Miss}$ value. Of course, this would not happen if we had a perfect detector. The second process, $pp \rightarrow Z \rightarrow \mu^+\mu^-$ can fake our signal of one muon and missing transverse energy, if one of the decay muons is not detected, i.e. leaves the detector unseen. In this case, also only one muon would be reconstructed in data and the second muon, which is not detected, would yield a missing transverse energy. Several observables can be used to separate the signal from the two mentioned background processes. Four promising reconstructed observables are shown in Figure 19, namely, the transverse momentum of the muon ($vecMuon\_PT$), the missing transverse energy of the

event ($fMET\_PT$), the isolation variable of the muon ($vecMuon\_TrkIso$03), as well as the pseudo-rapidity $\eta$ of the muon ($vecMuon\_Eta$) for all MC simulated events, which have exactly one reconstructed muon. The distributions for $Z \rightarrow \mu^+\mu^-$ are similar to the distribution of $W \rightarrow \mu\nu$ implying that this might be 'irreducible' background. However, we see significant differences in the multi-jet processes, which tend to have little missing transverse energy, low muon transverse momenta and large isolation variables. A first guess on possible selection criteria is therefore: $p_T(Muon) > 25$ GeV, $E_T^{Miss} > 30$ GeV and $\sum p_T^{iso}/p_T(Muon) < 0.1$.

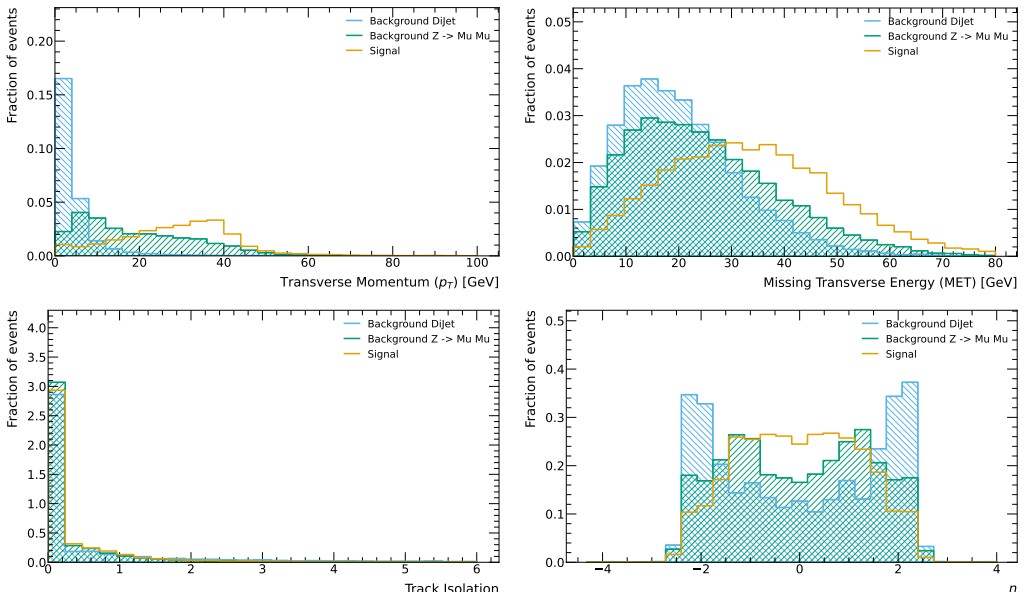

Figure 19: Comparison of $p_T$ Muon (top-left), ETMiss (top-right), Isolation (bottom-left), and $\eta$ (bottom-right) for signal and background MC Processes. The area of all distributions are normalized to unity.

Once the signal selection is defined, it is applied on all relevant MC simulated samples as well as data. The different simulated processes are then weighted according to their predicted probabilities and adjusted to the recorded size of the data-set. The distributions of MC signal and background processes are then added and compared to the observed data distributions. For example, the observed transverse momentum distribution of the reconstructed muon as well as the missing transverse energy of the selected events are shown in Figure 20. The good agreement between the prediction and the measurement illustrated the power the Standard Model of particle physics. Clearly, full physics analyses are significantly more complex, however, the basic concepts have been illustrated.

## 6 Machine Learning Tasks

Machine learning applications in experimental collider physics span a wide range of tasks: from classifying different physics processes, to reconstructing observables, simulating proton-proton collisions, and detecting anomalies in data that might hint at new physics beyond our current understanding. Below, we highlight a few illustrative tasks that demonstrate these concepts and show the field's diversity.

One central challenge in collider experiments is identifying different physical processes based solely on the particles observed in the detector. For example, consider the case of top-quark pair production. The top quark, the heaviest known elementary particle with a mass of

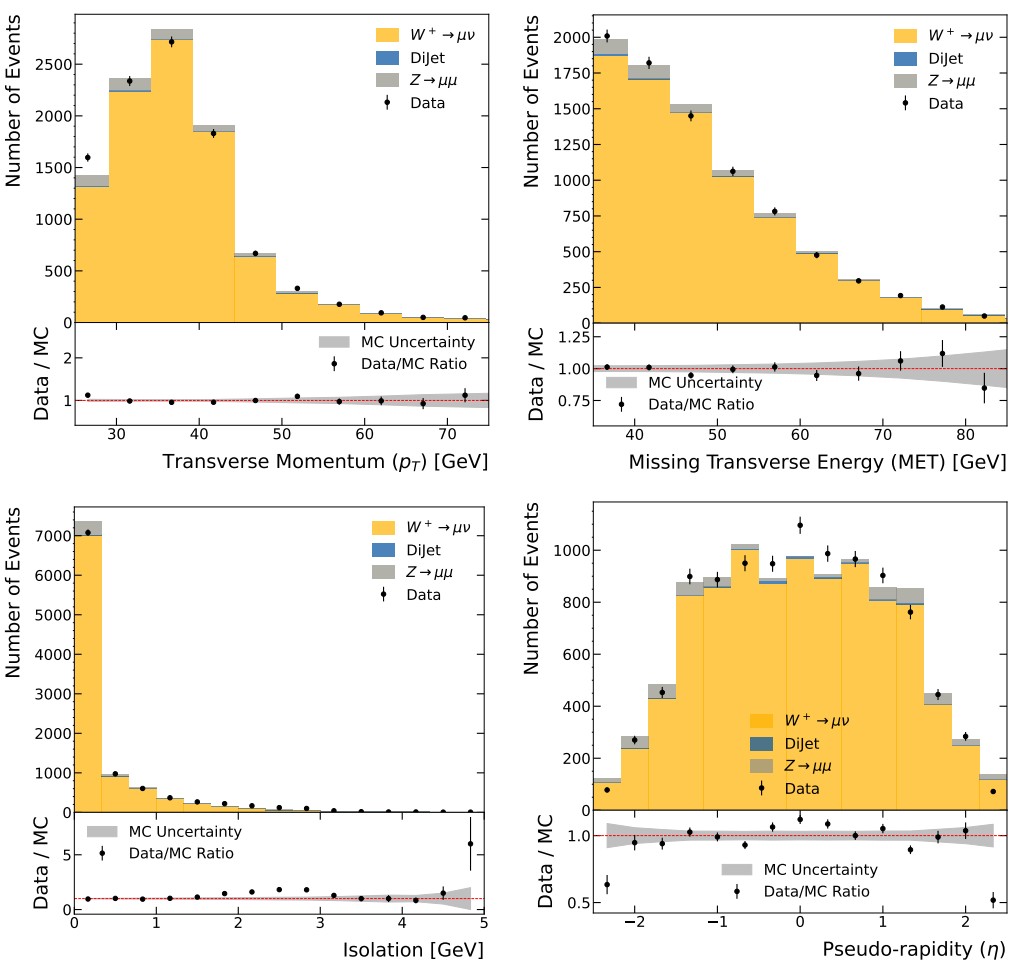

Figure 20: Top-left: Data/MC comparison for $p_T$ muon, top-right: Data/MC comparison for ETMiss, bottom-left: Data/MC comparison for Isolation, bottom-right: Data/MC comparison for $\eta$.

173 GeV, is nearly always produced in pairs and decays almost instantly into two b-quarks and two W bosons. These, in turn, often decay into lighter quarks, resulting in a final state of six jets of particles in the detector, two of which typically have displaced vertices (due to the b-quarks, as discussed in Section 2.5). However, the Standard Model also allows for other processes — such as the direct production of two c-quarks and two W bosons — that lead to similar signatures. This overlap makes it impossible to determine from a single event whether top-quarks were involved. By collecting a large amount of such events and analyzing differences in jet energies and angular distributions, we can statistically distinguish between these processes. A simple feedforward neural network can be trained on Monte Carlo (MC) simulations to take the kinematic properties of the six jets, along with displacement information, and output a score indicating the likelihood of a top-quark event. Once trained, this network can be applied to real data to perform signal-versus-background classification.

Another crucial task is the reconstruction of observables that are not directly measurable. A key example is the missing transverse energy observable, $\not{E}_T$, introduced in Section 2.5. This variable is sensitive to particles like neutrinos that do not interact with the detector. A basic $\not{E}_T$ reconstruction sums the transverse momentum of all observed particles, but this can be refined by excluding noise contributions (e.g., from pile-up interactions) or by adjusting for poorly modeled detector regions. Rather than manually crafting such corrections, a neural

network can be trained to regress the true $\not{E}_T$ based on all observable particle flow objects. Here too, MC simulation provides the ground truth for training.

A growing frontier involves generative models, such as generative adversarial networks (GANs) and normalizing flows, that can produce synthetic proton-proton collision events. These models aim to complement the expensive full-simulation pipelines used in HEP, offering event samples at a fraction of the computational cost. Other ML approaches seek to improve detector performance itself, for instance by denoising signals in calorimeters or refining the reconstruction of particle trajectories from detector hits, often using graph neural networks that naturally fit the irregular detector geometries.

Additionally, anomaly detection is an active research area, where unsupervised or weakly supervised ML models are trained on known Standard Model processes and then used to flag events that deviate from expected patterns. Such techniques have the potential to discover entirely new physics phenomena without requiring an explicit signal hypothesis in advance.

Beyond the diversity of tasks, the application of machine learning techniques to collider physics presents unique challenges and opportunities. The LHC produces some of the largest scientific datasets in the world, requiring scalable high-throughput algorithms. The data itself is structured in complex ways, often prompting innovations in neural architectures that embed physical symmetries (such as Lorentz invariance) or process irregular, sparse data (e.g., via set-based or graph-based models). The field also benefits from exceptionally accurate simulators, which enable supervised learning with high-quality labels. Another frontier is the deployment of ML algorithms within hardware-based trigger systems, which must process up to 40 million proton-proton collisions per second in real-time. These extreme latency and resource constraints motivate active research into network compression and acceleration techniques — such as pruning, quantization, knowledge distillation, and architecture search — to reduce the size and computational complexity of deep neural networks while preserving their predictive performance. These efforts not only enable real-time inference on specialized hardware like FPGAs and ASICs but also contribute broadly applicable advances in efficient deep learning, benefiting the wider ML community.

This landscape offers many avenues for computer scientists eager to contribute to fundamental physics using cutting-edge machine learning. For a more comprehensive overview of the current research frontier, we recommend the HEPML Living Review [15], which maintains an up-to-date catalog of developments in this vibrant and rapidly evolving field.

## 7   Conclusion

Our intention is to encourage an efficient transfer of the latest developments in the context of computer science to the field of fundamental physics. The primary objective of this paper was therefore to provide an introduction to the data collected at the Large Hadron Collider for computer scientists, allowing for a foundational platform for prospective interdisciplinary collaborations.

Moreover, we transformed the publicly available data from the Large Hadron Collider, which was initially stored in the ROOT data format—widely employed in high-energy physics—into Pandas dataframes, a format well-recognized in the realm of computer science. We hope that this significantly lowers the barrier of entry for future computer scientists at all levels to join the effort of unrevealing the secrets of the universe.

# Acknowledgements

This work has been conducted in the context of the AISafety Project, funded by the BMBF under the grant proposal 05D23UM1. .

**Funding information**    This work is funded by the BMBF under the grant proposal 05D23UM1.

# A    Transformation of CMS Open Data to Panda Data Frames

## A.1    Pandas Library

The python library **pandas** [16, 17] was designed with the goal in mind to bridge the gap between python and more domain-specific statistical and data analytical languages such as R. pandas is built on top of the NumPy library, enabling the use of fast and efficient methods to work with scientific data. In order to bridge the gap between python and other languages such as R, the creators of pandas initially provided two new structured data sets, **Series** for one-dimensional data and **DataFrames** for higher-dimensional data. In the context of this paper, the more interesting structure of data sets are the DataFrames, which were inspired by the R specific data.frame class. On top of replicating most functionalities of R's data.frame class, pandas DataFrames introduce enhancements such as automatic data alignment and hierarchical indexing. DataFrames are flexible in size and can be used to store mixed-type data as collections of columns, each column usually identified by a label. The indexing procedures introduced by DataFrames can be used to efficiently index over rows and columns. Moreover, the pandas library provides efficient ways to read and store DataFrames from and to memory.

On top of pandas efficient data sets, pandas also has a very active community, constantly improving the open-source project and enhancing it with new functionality. As such, the pandas library has cemented its place in many data scientific fields such as statistical analysis, financial analysis, and machine learning. Due to this popularity, a vast amount of documentation and additional resources, such as tutorials and usage examples, exist.

Therefore, we argue that the transformation of the CMS Open Data to pandas DataFrames not only provides an efficient alternative to storing high energy physics data, additionally, due to the library's steadfast presence in the current machine learning environment and its vast and engaged community, storing the data as pandas DataFrames enables a plethora of scientists not familiar with the ROOT file format, such as computer scientists, to use these high energy physics data in new analyses and deep learning models.

## A.2    Transformation Pipeline

The pipeline for transforming unfiltered ROOT files, available through the **CERN Open Data** platform [18], into filtered pandas DataFrames is implemented within a custom CMSSW environment (version CMSSW_5_3_32). For this study, CMS Open Data [19] from the 2011-2012 LHC run were used. The environment is configured to handle both AOD and AODSIM data formats, ensuring compatibility with the available datasets.

The pipeline begins by reading a series of ROOT file paths listed in .txt files from the CMS Open Data portal. Each .txt file contains multiple lines, with each line corresponding to a specific ROOT file stored on the CERN storage system. The ROOT files are then processed iteratively, one by one, within the CMSSW framework.

The next step involves filtering the data using a C++ EDAnalyzer within the CMSSW environment. This analyzer extracts a predefined set of variables from the events in each ROOT file. These variables, which typically include around 100 different quantities such as energies,

momenta, and particle IDs, are written to a new ROOT file. The filtered variables are stored in a ROOT TTree, a data structure that organizes the event-level information in a manner suitable for further processing.

Once the filtered ROOT files are generated, a separate Python script is employed to read them. The uproot library is used to access the TTree structure and extract the branches corresponding to the selected variables. These branches are then converted into NumPy arrays, which are subsequently placed into pandas DataFrames. This transformation involves a significant change in the data structure: the data, originally stored on a per-variable basis, is restructured into a per-event format (shown in Figure 21). This reorganization is necessary to align the data with the typical format required for machine learning applications, where each row represents a single event and each column represents a variable associated with that event.

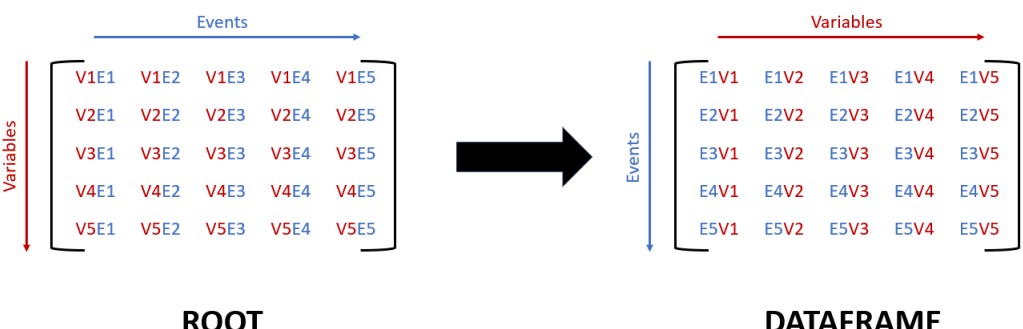

**ROOT**  **DATAFRAME**

Figure 21: Change of ordering of data from a per variable (red, V) basis, to a per event (blue, E) basis during the transformation of ROOT TTrees to pandas DataFrames.

After the transformation, the resulting pandas DataFrames are saved to disk using the Feather file format, chosen for its high performance across various use cases, as discussed in Section A.3. By default, the intermediary filtered ROOT files are deleted after being transformed into DataFrames, though it is possible to retain these files by adjusting a flag within the pipeline configuration.

Both the transformed DataFrames and, if retained, the filtered ROOT files are saved to a mounted directory, making them accessible on the local machine for subsequent analysis. A GitHub repository containing detailed instructions for setting up and running this pipeline, along with the necessary code, is available at [20].

## A.3 Bench-marking

pandas DataFrames can be saved in a variety of different file formats using different compression method, each combination of file format and compression method providing different advantages and disadvantages. The following section seeks to determine a combination, which - for the given data - presents the best overall performance. To this end, multiple datasets are benchmarked on three different tasks, namely the disk space required to save them, as well as their read and write speed. As a reference, the average disk usage of the initial filtered ROOT files is stated. Firstly, the most commonly used file formats in the context of pandas DataFrames were tested, using their default compression methods. The results can be seen in Figures 22 and 23. Each of the datasets used in this benchmark contains 10000 events.

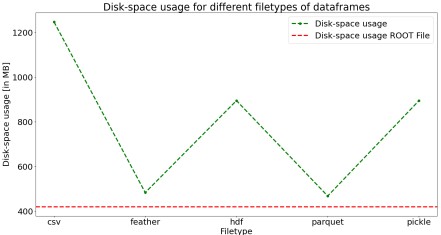

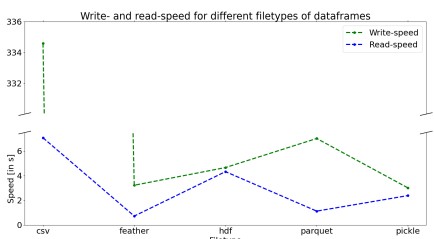

Figure 22: Average memory usage of the given file format over four different datasets.

Figure 23: Average read and write speed of the given file format over four different datasets.

When using default compression methods, each of the five file formats requires more disk space for saving the 10000 events as opposed to the ROOT files. The best performing file format on this disk usage benchmark is parquet, shortly followed by feather. On the read-speed benchmark, the feather file format is the best performing one, followed by parquet and pickle. As the DataFrames are only written once after being transformed from ROOT file to DataFrame, their write speed is not as important to us as their performance on the aforementioned benchmarks of disk-space and read-speed. All of the tested file formats are relatively close in performance on the write speed, except for the csv file format, which is significantly worse. When considering especially the performance on the disk usage and the read speed benchmarks, the two best file formats for our purposes appear to be feather and parquet. However, the disk usage required to save both the feather and the parquet files is still on average 10 to 15 percent larger than the disk usage required when saving the filtered ROOT files. Hence, in order to combat this issue while ideally also keeping the read and write speed as low as possible, different compression methods for the feather and parquet file formats were tested. The resulting benchmarks can be seen in Figures 24 and 25.

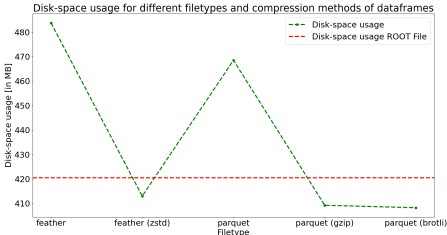

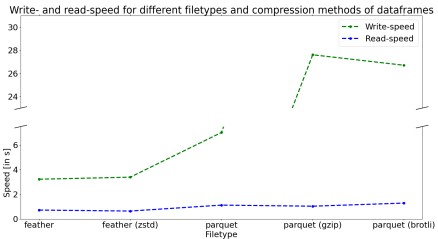

Figure 24: Average memory usage of the given file format and compression method over four different datasets.

Figure 25: Average read and write speed of the given file format and compression method over four different datasets.

The feather file format using zstd compression, as well as parquet using gzip and brotli compressions manage to outperform even the initial ROOT files on the benchmark of disk space requirement, with the parquet compressions requiring slightly less disk-space than the feather zstd compression. However, for the write and read speed benchmarks using these compression methods, the feather file format compressed using zstd outperforms all parquet compression methods on both benchmarks, especially on the write speed benchmark. Another aspect to be considered is the compatibility of the discussed file formats. In this regard, the csv files offer the largest compatibility, as they are basically universally compatible. The feather and parquet file formats are both based on the Apache Arrow format, hence they are only compatible with languages that support the Apache Arrow, namely C, C++, Go, Java, JavaScript, Julia, Python,

1166 R, Ruby, and Rust.

1167     After considering the benchmarking results gathered, the decision was made to save the
1168 resulting DataFrames as feather files, using the zstd compression, as this method appears to of-
1169 fer the best overall performance on the data at hand, as well as providing decent compatibility
1170 with many popular programming languages in the context of data science and deep learning.

## A.4   Tables

| Amount of Objects encountered and their Energy for Top-Top Jets | | |
|---|---|---|
| **Object** | **Mean Amount Encountered per Event** | **90% Energy Range, $p_T$ [GeV]** |
| Muon | 2.5582 | 0.8767 - 73.5747 |
| Electron | 17.4122 | 2.9442 - 88.9196 |
| Vertex | 1.6322 | - |
| Tauon | 78.2536 | 0.8626 - 21.9836 |
| Photon | 2.0526 | 10.3277 - 91.0337 |
| McTruth | 664.3001 | 0.0610 - 19.1225 |
| Jets | 78.2536 | 3.3669 - 37.9243 |
| Particle Flow | 1361.7938 | 0.1002 - 2.1753 |

Table 15: Mean amount of objects encountered per event and their 90% energy intervals for Top-Top Jets.

| Amount of Objects encountered and their Energy for WW Jets | | |
|---|---|---|
| **Object** | **Mean Amount Encountered per Event** | **90% Energy Range, $p_T$ [GeV]** |
| Muon | 1.4928 | 0.8533 - 74.6767 |
| Electron | 15.2397 | 3.0247 - 90.3210 |
| Vertex | 0.9016 | - |
| Tauon | 71.1567 | 0.8094 - 16.3372 |
| Photon | 0.9956 | 10.8430 - 93.2002 |
| McTruth | 560.8589 | 0.0542 - 5.9356 |
| Jets | 71.1567 | 3.3061 - 24.2140 |
| Particle Flow | 1165.6494 | 0.0889 - 1.7910 |

Table 16: Mean amount of objects encountered per event and their 90% energy intervals for WW Jets.

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
