# Peer review of "Introduction to the Usage of Open Data from the Large Hadron Collider for Computer Scientists in the Context of Machine"

_SciPost Physics Lecture Notes, doi:SciPost Phys. Lect. Notes 96 (2025)_

## Round 1 · Referee Report · Anonymous (Referee 1) · 2025-5-1

Strengths

  1. The paper gives a good introduction to LHC physics and data analysis. It is both succinct and generally provides sufficient detail for both the physics background and the experimental aspects relevant for data analysis

  2. It is accompanied by an open data set, which has been processed into a data format broadly used beyond particle physics. Such a dataset will be useful for those outside of the field to get an introduction to what particle physics data looks like and develop some basic applications.

Report

I recommend the paper for publication. I have several comments which could improve the quality of the text which should be addressed first.

Requested changes

My largest comment is that given the target audience of computer scientists being introduced to particle physics, more could be done to excite them about the interesting and unique challenges and applications of machine learning to LHC physics.

I highlight a few areas this could be emphasized, in addition to other minor comments to improve the clarity of the text below.

Comments:

Early in the introduction perhaps it would be good to briefly overview in two or sentences what the central aims of particle physics research are, and also why machine learning has become so prevelant (large complex datasets). This introduction should try to excite computer scientists that particle physics would be an interesting domain for them to work

L43 The last two sentences of this introductory paragraph feel out of place. As this off handed references to these specific application of ML would not be easy for a computer scientist to follow (ie they would not know what 'event reconstruction' is )

If the intended audience is computer scientists it is perhaps not good to use phrases like "new physics beyond the standard model" without proper explanation

L108: The phrase 'while this may seem like science fiction' seems a little out of place

L119: I don't think the phrase 'infinite distances' should be used, perhaps can be rephrased to large / macroscopic distances or something similar

L339 " to estimate the origin of the particle, one needs to assume its origin, e.g. the primary vertex" This sentence is circular, primary vertex has also not been explained

L435: 0.937 GeV should be 0.938, matching previous line and measured value

L660 Perhaps it would be better to phrase as 'several approximations need to made' rather than 'assumptions' (also later in the paragraph). The standard model itself is an assumption

L734: 'reassemble' -> 'resemble' perhaps?

Table 5: Some of the descriptors could be improved. Eg degrees of freedom or chi2 should mention this is from the track fit as this is not obvious to non-experts (also Table 9)

Sec 4.2.5 I find the description of the saved vertex objects very confusing. The attempt to relate the experimental vertices to vertices in Feynman diagrams seems very tenuous. The vertices we measure are just from the origin of tracks. Except in the case of long lived particles all the particles from the hard interaction will originate from this point, not just the ones associated to a certain vertex in the Feynman diagram. I suggest to stick to the 'experimental' view of vertices related to tracking.

Table 10: What is the "statistical fraction" of HoverE?

Table 11: the Id_1 and Id_2 variables are the flavor of the incident partons from each proton in the collision? the description 'first partons' is kind of unclear

Table 12: Explain Beta and Beta star variables in more detail. Should clarify whether the jet momenta have been corrected with the JEC's already or not

Table 13: Should clarify somewhere what 'type' of MET is being stored here (within CMS there are several)

Should clarify earlier in the introduction what dataset you are looking at (ie Run-1 CMS)

L922: Unsure what is meant by "since we cannot observe the W boson direction"

Table 14: Cross sections should be given for the MC processes and luminosities for the data samples. Approximate sizes for your pandas dataset of each process would be very helpful to the reader as well

Section 5: Without the aforementioned cross sections and luminosities, it would not be possible for the reader to follow along and implement this example. Additionally, as this is meant to be a pedagogical example it would be strongly advised to include a public code release which implements the selections and makes the plots shown in the paper

L969: "over the reconstruction of observables" is awkward phrasing

Table 14: Please include proper citations to the CMS datasets in the references in addition to the DOI. The webpage of each page has a description of how CMS would like it to be cited. Such a citation is important to give credit to CMS for releasing their data and encourage them to continue to do so in the future

Section 6:
The two examples given are good first examples of the application of ML in HEP, however more could be done to excite the reader about further possibilities.
For example unique challenges and opportunities of the application of ML to particle physics could be discussed, such as some the largest scientific datasets, unique data formats leading to innovation of novel ML architectures imposing physical symmetries, high quality simulators available for training purposes, strict latency requirements for 'online' ML algorithms deployed in the triggers leading to innovations in network compression, etc.
A short overview of the diverse range of ML+LHC applications under current research (a few sentences) could help excite CS researchers about the possibilities
Then a citation of a recent review of ML in HEP, or additionally the HEPML living review webpage, could be given so the interested reader somewhere to look if they wanted to more ideas of ML applications.

Appendix A.2: Some more technical details should be given as to the stages of the CMS processing for documentation about the pipeline, Eg that you starting from the AOD data format

Recommendation

Ask for minor revision

---

## Round 2 · Referee Report · Anonymous (Referee 1) · 2025-5-19

Strengths

The paper gives a good introduction to LHC physics and data analysis. It is both succinct and generally provides sufficient detail for both the physics background and the experimental aspects relevant for data analysis

It is accompanied by an open data set, which has been processed into a data format broadly used beyond particle physics. Such a dataset will be useful for those outside of the field to get an introduction to what particle physics data looks like and develop some basic applications.

Report

The authors have addressed all my comments from the previous review . I now recommend publication.

Recommendation

Publish (meets expectations and criteria for this Journal)

---

## Round 2 · Author Response

Dear Editorial Team and Referee,

We would like to thank you for your feedback on our submission "Introduction to the Usage of Open Data from the Large Hadron Collider for Computer Scientists in the Context of Machine Learning" to the SciPost Physics Lecture Notes. We appreciate the opportunity to revise and resubmit and have considered the comments, resulting in the changes made detailed below. The format for these changes will be stating the review, followed by the changes made prompted by the respective comment.

---

## Round 2 · List of Changes

1. Early in the introduction perhaps it would be good to briefly overview in two or sentences what the central aims of particle physics research are, and also why machine learning has become so prevelant (large complex datasets). This introduction should try to excite computer scientists that particle physics would be an interesting domain for them to work

Added two paragraphs further discussing and high-lighting why particle physics data is also interesting in the ML sphere:

  1. L43 The last two sentences of this introductory paragraph feel out of place. As this off handed references to these specific application of ML would not be easy for a computer scientist to follow (ie they would not know what 'event reconstruction' is )

Changed the entire introductory paragraph, briefly explaining concepts such as the Standard Model. Additionally simplified some concepts (by using non-physics terms to describe the processes).

  1. If the intended audience is computer scientists it is perhaps not good to use phrases like "new physics beyond the standard model" without proper explanation

See changes for review 2

  1. L108: The phrase 'while this may seem like science fiction' seems a little out of place

Removed it

  1. L119: I don't think the phrase 'infinite distances' should be used, perhaps can be rephrased to large / macroscopic distances or something similar

rephrased to “distances on a cosmic scale”

  1. L339 " to estimate the origin of the particle, one needs to assume its origin, e.g. the primary vertex" This sentence is circular, primary vertex has also not been explained

Changed the part to: … and to determine the particle’s trajectory and origin point, it is typically assumed that the particle originated from a specific reference point in the collision, known as the primary vertex—the location where the initial proton-proton collision occurs.

  1. L435: 0.937 GeV should be 0.938, matching previous line and measured value

Fixed it

  1. L660 Perhaps it would be better to phrase as 'several approximations need to made' rather than 'assumptions' (also later in the paragraph). The standard model itself is an assumption

Rephrased to approximations

  1. L734: 'reassemble' -> 'resemble' perhaps?

made it “resemble”

  1. Table 5: Some of the descriptors could be improved. Eg degrees of freedom or chi2 should mention this is from the track fit as this is not obvious to non-experts (also Table 9)

Adjusted the descriptions to be more informative and direct

  1. Sec 4.2.5 I find the description of the saved vertex objects very confusing. The attempt to relate the experimental vertices to vertices in Feynman diagrams seems very tenuous. The vertices we measure are just from the origin of tracks. Except in the case of long lived particles all the particles from the hard interaction will originate from this point, not just the ones associated to a certain vertex in the Feynman diagram. I suggest to stick to the 'experimental' view of vertices related to tracking

Completely re-wrote the section to factor in the remarks

  1. Table 10: What is the "statistical fraction" of HoverE?

rephrased to just fraction

  1. Table 11: the Id_1 and Id_2 variables are the flavor of the incident partons from each proton in the collision? the description 'first partons' is kind of unclear

Adjusted the descriptions, indeed they are the incident partons from each proton (Id_1 = incident parton from first proton beam, Id_2 = incident parton from second proton beam)

  1. Table 12: Explain Beta and Beta star variables in more detail. Should clarify whether the jet momenta have been corrected with the JEC's already or not

Further explained beta and beta star, clarified that the JEC's have not yet been applied.

  1. Table 13: Should clarify somewhere what 'type' of MET is being stored here (within CMS there are several)

Clarified/Added it (we use only the particle flow definition of MET)

  1. Should clarify earlier in the introduction what dataset you are looking at (ie Run-1 CMS)

Added this sentence to the introduction: … The datasets used in this work are based on Run 1 CMS data collected in 2012, which provide a rich and well-documented benchmark for developing and testing machine learning models. …

  1. L922: Unsure what is meant by "since we cannot observe the W boson direction"

Sorry, this was completely out of place and we changed that.

  1. Table 14: Cross sections should be given for the MC processes and luminosities for the data samples. Approximate sizes for your pandas dataset of each process would be very helpful to the reader as well

Added a column for cross-sections / luminosities, and also added column for how many samples we have.

  1. Section 5: Without the aforementioned cross sections and luminosities, it would not be possible for the reader to follow along and implement this example. Additionally, as this is meant to be a pedagogical example it would be strongly advised to include a public code release which implements the selections and makes the plots shown in the paper

Added explicit references in the introduction of the section to the data (Hugging Face) and the Code (GitHub)

  1. L969: "over the reconstruction of observables" is awkward phrasing

Rephrased the entire paragraph as: Machine learning applications in experimental collider physics range from classifying different physics processes, to reconstructing observables and simulating proton-proton collisions, and to detecting anomalies in data that might hint at new physics beyond our current understanding. The following two examples illustrate these basic concepts.

  1. Table 14: Please include proper citations to the CMS datasets in the references in addition to the DOI. The webpage of each page has a description of how CMS would like it to be cited. Such a citation is important to give credit to CMS for releasing their data and encourage them to continue to do so in the future

Included proper citations from the Open Data Portal

  1. Section 6: The two examples given are good first examples of the application of ML in HEP, however more could be done to excite the reader about further possibilities. For example unique challenges and opportunities of the application of ML to particle physics could be discussed, such as some the largest scientific datasets, unique data formats leading to innovation of novel ML architectures imposing physical symmetries, high quality simulators available for training purposes, strict latency requirements for 'online' ML algorithms deployed in the triggers leading to innovations in network compression, etc. A short overview of the diverse range of ML+LHC applications under current research (a few sentences) could help excite CS researchers about the possibilities Then a citation of a recent review of ML in HEP, or additionally the HEPML living review webpage, could be given so the interested reader somewhere to look if they wanted to more ideas of ML applications.

Completely re-wrote the entire section 6, including additional and more descriptive examples / challenges, as well as a reference to the living review HEPML webpage.

  1. Appendix A.2: Some more technical details should be given as to the stages of the CMS processing for documentation about the pipeline, Eg that you starting from the AOD data format

Adjusted the entire section, hopefully providing a more technical inside into the process of transforming the CMS Open Data to pandas DataFrames, from start to finish.

---

## Editorial Decision

published